# Microscopic structure of the polymer-induced liquid precursor for calcium carbonate

Yifei Xu[1,2], Koen C.H. Tijssen[3], Paul H.H. Bomans[1,2], Anat Akiva[1,2], Heiner Friedrich[1,2], Arno P.M. Kentgens[3] & Nico A.J.M. Sommerdijk [1,2]

Many biomineral crystals form complex non-equilibrium shapes, often via transient amorphous precursors. Also in vitro crystals can be grown with non-equilibrium morphologies, such as thin films or nanorods. In many cases this involves charged polymeric additives that form a polymer-induced liquid precursor (PILP). Here, we investigate the $CaCO_3$ based PILP process with a variety of techniques including cryoTEM and NMR. The initial products are 30–50 nm amorphous calcium carbonate (ACC) nanoparticles with ~2 nm nanoparticulate texture. We show the polymers strongly interact with ACC in the early stages, and become excluded during crystallization, with no liquid–liquid phase separation detected during the process. Our results suggest that "PILP" is actually a polymer-driven assembly of ACC clusters, and that its liquid-like behavior at the macroscopic level is due to the small size and surface properties of the assemblies. We propose that a similar biopolymer-stabilized nanogranular phase may be active in biomineralization.

[1] Department of Chemical Engineering and Chemistry, Laboratory of Materials and Interface Chemistry and Centre for Multiscale Electron Microscopy, Eindhoven University of Technology, PO Box 513, 5600 MB Eindhoven, The Netherlands. [2] Institute for Complex Molecular Systems, Eindhoven University of Technology, PO Box 513, 5600 MB Eindhoven, The Netherlands. [3] Solid-state NMR Group, Institute for Molecules and Materials, Radboud University Nijmegen, Heyendaalseweg 135, 6525 AJ Nijmegen, The Netherlands. Correspondence and requests for materials should be addressed to A.P.M.K. (email: A.Kentgens@nmr.ru.nl) or to N.A.J.M.S. (email: n.sommerdijk@tue.nl)

In biominerals such as bone or the nacre of seashells, the formation of mineral crystals is commonly guided by highly charged biopolymers[1–4]. In spite of their intrinsic crystallographic symmetries, they are frequently molded into complex non-equilibrium shapes[5], often through the deposition and transformation of amorphous precursor phases[6–10]. A similar level of control over crystal morphology can be realized in vitro, also through the interaction with charged (bio)polymers for inorganic[11–15] as well as organic[16] materials, which was most extensively demonstrated for $CaCO_3$[17–23]. In the latter case, the presence of charged polymers such as poly(aspartic acid) (pAsp)[18], poly(acrylic acid) (pAA)[17,24], poly(allylamine hydrochloride) (pAH)[19], and double-stranded DNA (ds-DNA)[23] can stabilize amorphous calcium carbonate (ACC)[25], and under specific conditions lead to the formation of a liquid-like precursor, which has been explored in the formation of crystals with non-equilibrium morphologies[18–22]. It was reported that droplets (Ø 100 nm–5 μm) of this so-called polymer-induced liquid precursor (PILP)[26] can coalesce and form thin films on solid substrates[18–20], or infiltrate into nanopores[20,21], where they transform into solid ACC and subsequently crystallize to calcite or vaterite[18], with their morphologies preserved. The formation of these non-equilibrium morphologies have been attributed to the liquid-like nature of PILP, being able to wet the solid substrates, or to be capillarilly absorbed into the nanopores[18,21].

PILP was first reported for $CaCO_3$ based on in situ optical microscopy (OM)[18], however in later papers, the existence of PILP (and thereby its liquid nature) was mainly inferred from static observations of dried samples, using atomic force (AFM)[27], scanning electron (SEM)[21,28], or transmission electron microscopy (TEM)[22]. Direct evidence for the liquidity of PILP could only be obtained by characterization in hydrated status, such as cryoTEM or liquid-state nuclear magnetic resonance (NMR) spectroscopy. The PILP droplets are expected to be continuous objects with smooth edges under cryoTEM[19,29,30]. Furthermore, the liquid–liquid phase separation should be reflected by extra peaks in liquid-state NMR measurements[31]. The first nanoscale identification of PILP in solution waited until 2012 when Cantaert et al.[19] used cryoTEM to indicate the deformability of the droplets, and Bewernitz et al.[32] used $^{13}C$ NMR spectroscopy to demonstrate the existence of a $CaCO_3$ component with a $T_2$ relaxation time and self-diffusion coefficient in agreement with a liquid phase.

Nevertheless, several experimental observations cannot be explained by the current PILP droplet model. For example, it was found that PILP droplets show a gel-like elasticity[33]. And in contrast to conventional liquids that usually show intermediate contact angles, PILP shows either complete wetting (contact angle = 0°) or non-wetting (contact angle >150°) behavior on different substrates[34]. Moreover, the $CaCO_3$ thin films resulting from the PILP process appear to consist of aggregated ~100 nm-sized ACC nanoparticles (NPs)[27,35], where products with smooth surfaces and a homogeneous interior should be generated, if PILP indeed consists of liquid droplets.

The PILP process has also been suggested to have its equivalents in biomineralization[26], where charged polymers are involved in controlling the crystallization processes[1–4], and similar non-equilibrium morphologies are generated[5]. Indeed, very similar ~100 nm nanogranular textures are also observed in many $CaCO_3$ biominerals such as nacre of seashells[36] or the sea urchin spine[37,38], and polypeptides with charge densities close to biomineralization proteins are able to generate PILP phases[20]. The formation of colloidally stabilized nanogranular calcium phosphate globules using charged polypeptides has also been instrumental in the in vitro infiltration of collagen fibrils with apatite nanocrystals[39], where in biology similar processes may be active. Hence, beyond the importance for achieving control over the morphology of crystalline materials, understanding the PILP process may lead to mechanistic insights into the formation processes of biominerals.

Here we study the formation and transformation processes of PILP using multiple characterization techniques, including cryoTEM and NMR. By vitrifying liquid samples collected at different growth stages, cryoTEM uniquely allows us to monitor the evolution of products in their native hydrated state and with nanometer resolution[40]. We focus on several $CaCO_3$ based systems including the archetypal and most common one using pAsp. Through comparing the experimental observations with those obtained for other known systems such as ds-DNA, pAA and pAH, we demonstrate the generality of the process. The use of ds-DNA especially benefits the study, as the diameter of ds-DNA (2.4 nm) allows the cryoTEM visualization of the biopolymer during the mineralization process[41], and phosphate groups in the DNA backbone allow the detection of phosphorous signals in NMR[42,43].

The combination of two-dimensional (2D) and three-dimensional (3D) high-resolution cryoTEM with Fourier transform infrared (FTIR), pH, and $[Ca^{2+}]$ measurements, dynamic light scattering (DLS)/zeta potential measurements, inductively coupled plasma optical emission spectrometry (ICP-OES) as well as liquid- and solid-state NMR spectroscopy provides unique insights into the microscopic structure of PILP, indicating that they are actually 30–50 nm ACC NPs with ~2 nm nanoparticulate texture. The results suggest that PILP is actually a polymer-driven assembly of ACC clusters, and that its liquid-like behavior at the macroscopic level is due to the small size and surface properties of the assemblies.

## Results

**SEM and OM visualization of the PILP process.** $CaCO_3$/poly-(α,β)-DL-aspartic acid (pAsp, mw = 2000–11000 g mol⁻¹) dispersions were prepared using literature procedures (setup shown in Supplementary Fig. 1a), and shown to exhibit the crystallization behavior typical for PILP systems[18]. On hydrophilic glass slides this yielded—as expected—amorphous films (Fig. 1a) extending over millimeters, that in time transformed into crystalline platelets of calcite or vaterite (Fig. 1b, c, see also Supplementary Fig. 2 for Raman spectroscopy), and occasionally also into their 3D counterparts (insets of Fig.1b, c)[18]. Under the same experimental conditions, ~20 μm-sized calcite crystals were formed within 30 min without the presence of pAsp (Supplementary Fig. 3), showing the ability of pAsp to inhibit the crystallization and stabilize ACC phase. When track etch polycarbonate membranes were exposed to this dispersion, rod-like $CaCO_3$ crystals with concave tips were formed inside the 50 or 200 nm pores with presence of pAsp (Fig. 1d and inset, see also Supplementary Fig. 4 and Supplementary Table 1), consistent with previous reports of PILP growth systems[20,21]. Without the presence of pAsp, however, only small amount of short $CaCO_3$ nanorods are formed within the track etch membrane (Supplementary Fig. 4a and d), which shows crystal-facet on their tips (Supplementary Fig. 5).

In situ differential interference contrast optical microscopy (DICOM, setup shown in Supplementary Fig. 6) showed the formation of the amorphous film and its subsequent dissolution upon the nucleation and growth of crystalline platelets (Fig. 1e–h). In agreement with the original observations of Gower et al.[18], ~2 μm-sized moving particles were observed near the glass/solution interface during formation of the film (inset of Fig. 1e, see also Supplementary Fig. 7 and Supplementary Movie 1). However, it was never observed as proposed by Gower et al.[18] that these particles deposited and subsequently coalesced with the film.

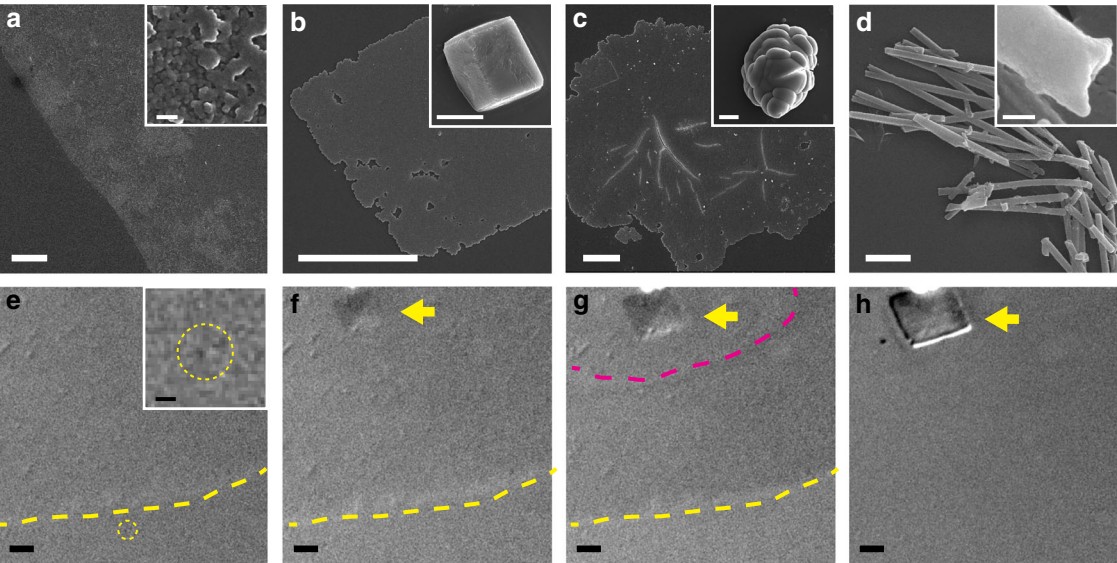

**Fig. 1** SEM and OM observations of the $CaCO_3$ thin film formation process with 25 mg $L^{-1}$ pAsp and 10 mM $CaCl_2$. **a–d** SEM images of the dried products. **a** Thin film formed on a glass slide at 5 h. Inset is a zoom-in image showing the ~60 nm-sized NPs in the film. **b, c** Calcite and vaterite platelets formed at 24 h, with insets showing 3D crystals also observed in the assay, respectively. **d** Rod-like $CaCO_3$ crystals formed within a track etch membrane with 200 nm-sized pores. Inset shows the concave tip of the nanorods. **e–h** Differential interference contrast optical microscopy (DICOM) in situ observation of the thin film formation process taken at 151, 248, 266, 335 min after reaction started, respectively. The thin film formed in **e**. A ~2 μm-moving particle is highlighted by the yellow circle, with its zoom-in image shown in the inset. A crystalline platelet nucleated within the film in **f**, and grew in **g** and **h**. The film near the growth front of the crystalline platelet started to dissolve in **g**. The dissolution extended outside to the film edge and the film was fully dissolved in **h**. The boundary of thin film is highlighted by yellow dashed line in **e–g**. The calcite thin platelet is highlighted by yellow arrows in **f–h**. The dissolution front of the film is highlighted by the magenta-dashed line in **g**. Scale bars: **d** and inset of **e**: 2 μm. Inset of **a** and **d**: 100 nm. Other images: 10 μm

## 2D and 3D CryoTEM visualization of the PILP process.

To further investigate the mechanism of thin film formation, samples were taken from the crystallization solution at different growth stages and visualized by cryoTEM. The earliest products were observed after ~150 min of growth, which are NPs with diameters of ~50 nm (Fig. 2a, see Supplementary Figs. 8, 9 for the cryoTEM images of even earlier stages). High-magnification images showed that the NPs appear to consist of assemblies of ~2 nm subunits (inset of Fig. 2a), similar to those observed in a pAA/ACC hydrogel using conventional (dry) TEM[44]. The NPs subsequently grew in size and aggregated to form larger structures, but did not coalesce to form continuous objects with smooth edges as would be expected for liquid droplets (Fig. 2b, see also Supplementary Fig. 10 for the expected morphology of liquid droplets visualized by cryoTEM). At ~250 min, the onset of film formation was observed to occur through further aggregation of NPs (Fig. 2c), which were still amorphous as demonstrated by selected area electron diffraction (SAED, inset of Fig. 2c). Also after freeze drying (see Supplementary Fig. 11), the morphology of the aggregates retained their granular appearance, similar to what has been observed previously with conventional TEM[35], indicating that they are formed by aggregation of the NPs rather than by their coalescence. The results clearly contrast with our previous cryoTEM study that used similar reaction conditions but without the presence of polymers[45], where ACC NPs without fine structure were formed, and crystallization happened within 5 min.

In addition to pAsp, also other polymers with charged functional groups such as pAA[17,24], pAH[19], polypeptides[20], and ds-DNA[23] have been shown to induce $CaCO_3$ mineralization behavior with all the characteristics (thin film formation, infiltration of porous substrates) of the PILP process. For comparison, thin films of $CaCO_3$ were grown using ds-DNA as the crystallization control agent, following published procedures, i.e., from a 10 mM $CaCl_2$ solution containing 2.5 g $L^{-1}$ of ds-DNA (300 bps, mw ≈ 225,000 g $mol^{-1}$)[23]. The ds-DNA used

(from salmon sperm, the same as in our original work) contained a small amount of Mg (15.25 mg $g^{-1}$, see ICP-OES results in Supplementary Table 2), leading to a $Mg^{2+}$ concentration of ~1.6 mM in the crystallization solution. This is expected not to significantly affect the results, since $Mg^{2+}$ only notably influences the mineralization of $CaCO_3$ at Mg: Ca concentration ratios ≥1[46].

Under the same experimental conditions as used for pAsp, ds-DNA also facilitated the growth of cylindrical crystals within nanopores (see Supplementary Fig. 4 and Supplementary Table 1), although it was less effective than pAsp. After 30–60 min, cryoTEM visualized NPs with ~30 nm diameter, which are near-identical to those formed in the presence of pAsp, and also appear to be assembled from ~2 nm subunits (Fig. 2d and inset). In this case however, also the macromolecules (ds-DNA, highlighted by yellow arrows in Fig. 2d) could be observed, intertwined with the ~2 nm subunits and protruding into the solution. Just like the NPs formed in the presence of pAsp, the ds-DNA-based NPs aggregated to form an amorphous thin film (Fig. 2e, f, see also Supplementary Fig. 12) that transformed into plate-like crystals typical for a PILP system. Also in this case, no signs of liquid-like coalescence were observed. Similar NPs with ~2 nm nanoparticulate textures were found in the early stages of the pAA or pAH (25 mg $L^{-1}$)-induced PILP processes (Supplementary Fig. 13), underlining the universality of our observations. It is noteworthy that the assemblies induced by different polymers are different in sizes (~200 nm-sized assemblies could be observed for pAH) and shapes (the particles induced by pAH or ds-DNA seem to be more spherical than those formed with pAsp or pAA), which may result from the different properties of the polymers (e.g., molar weight, charge density, length, and so on). Meanwhile, when pAH was used at a higher concentration (1000 mg $L^{-1}$), indeed droplet-like objects were observed as was originally reported by Cantaert et al.[19] (Supplementary Fig. 14).

Cryo-electron tomography (cryoET, 3D cryoTEM) was used to visualize the 3D structure of the NPs, showing that they are nearly

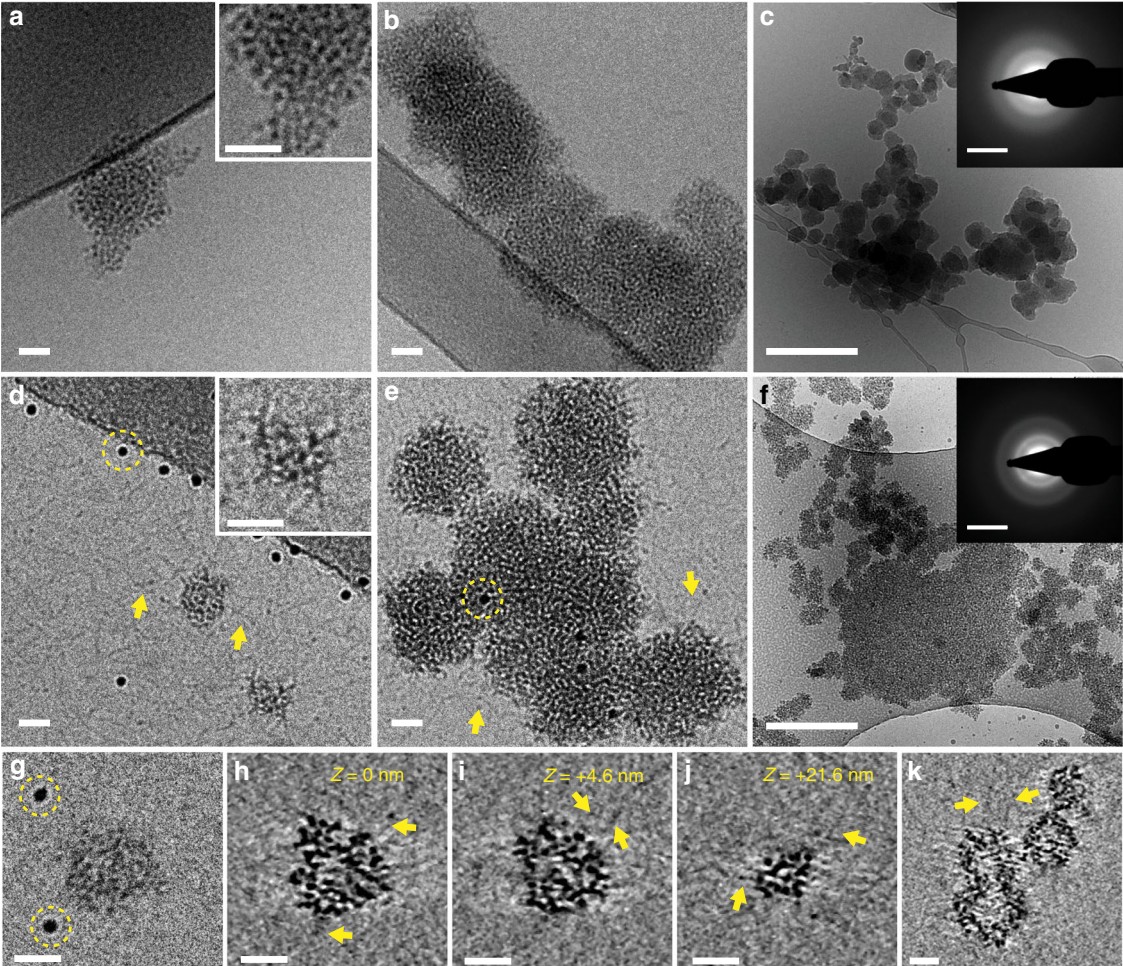

**Fig. 2** CryoTEM observations of CaCO$_3$ thin film formation process with 10 mM CaCl$_2$. **a–c** Images showing the evolution of product morphologies with 25 mg L$^{-1}$ pAsp at different growth stages. **a** ~60 nm-sized NP (150–200 min). The inset is a zoom-in image showing the ~2 nm-sized subunits. **b** Aggregated NPs (200–250 min). **c** Film aggregated by NPs (250–300 min), with SAED pattern in the inset showing it is amorphous. **d–f** Images of products grown with 2.5 g L$^{-1}$ ds-DNA. **d** Approximately 30 nm-sized NPs (30–60 min). The inset is a zoom-in image showing the ~2 nm-sized subunits. **e** Aggregated NPs (60–120 min). **f** Film aggregated by the NPs (~120 min), with SAED pattern in the inset showing it is amorphous. **g–k** Tomography of a NP grown with 2.5 g L$^{-1}$ of ds-DNA (30 min). **g** Tomo tilt series image at 0°. **h–j** Computer-generated cross-section slices with offsets of 0, 4.6, or 21.6 nm from the center of NP in Z-axis, respectively. **k** A cross-section slice of four NPs attached together (60 min), indicating they are hollow. **h–k** are median-filtered (filter size 3 × 3 × 3 pixels) to remove noise for better visibility. The gold markers for tomography are highlighted by yellow circles in **d**, **e**, and **g**. Some ds-DNA molecules are highlighted by yellow arrows in **d**, **e** and **h–k**. Scale bars: **c** and **f**: 500 nm. Inset of **c** and **f**: 5 nm$^{-1}$. Other images: 20 nm

spherical with an irregular, rough surface (Fig. 2g–j, see also Supplementary Movie 2). Ds-DNA molecules were observed protruding the surface of the particles (Fig. 2h, j, yellow arrows), but could not be observed in their interior due to a lack of contrast. Detailed morphological analysis (see Methods: morphological processing and analysis of tomographic results) indicated the NPs have a bicontinuous structure consisting of fused mineral subunits, combined with interpenetrating channels filled with solution and/or ds-DNA molecules that comprise ~50% of the particle volume (Supplementary Table 4). Surprisingly, tomography also revealed that some of the NPs are hollow (Fig. 2k, see also Supplementary Movie 3), which further indicates that they are not liquid droplets as those should have a homogenous and continuous structure.

**Monitoring the evolution of mineral phases**. To obtain insight into the compositional evolution of the mineral phases, a combination of different techniques was used, employing attenuated total reflection FTIR (ATR-FTIR) spectroscopy, pH and Ca$^{2+}$

concentration measurements, and DLS/zeta potential measurements as well as liquid- and solid-state NMR. To obtain homogeneous solution conditions for these measurements, slow stirring (100 r.p.m., setup shown in Supplementary Fig. 1b) was applied. CryoTEM revealed that under these conditions, the same composite NPs were formed as in the static experiments (Fig. 3a), confirming that the slow stirring did not significantly affect the PILP formation process, although thin film formation was prevented and instead a precipitate formed (Fig. 3b–d).

Double-stranded DNA can adopt different conformations depending on the base sequence, electrolyte composition and concentration, pH, temperature, and so on[42,47]. Although for longer nucleic acid fragments (>100 bps) the $^{31}$P NMR signals from individual residues cannot be resolved[42], the spectra still give insight into the occurrence of different ds-DNA conformations. In water solution, ds-DNA generally adopts B-conformation, while dry ds-DNA often adopts the A-conformation[48]. The $^{31}$P liquid-state NMR (Fig. 4a, b) of a neat 2.5 g L$^{-1}$ ds-DNA solution indeed showed a single peak at $\delta = -3.87$ p.p.m. (peak width 0.60 p.p.m.), which corresponds to B-DNA[42]. After mixing

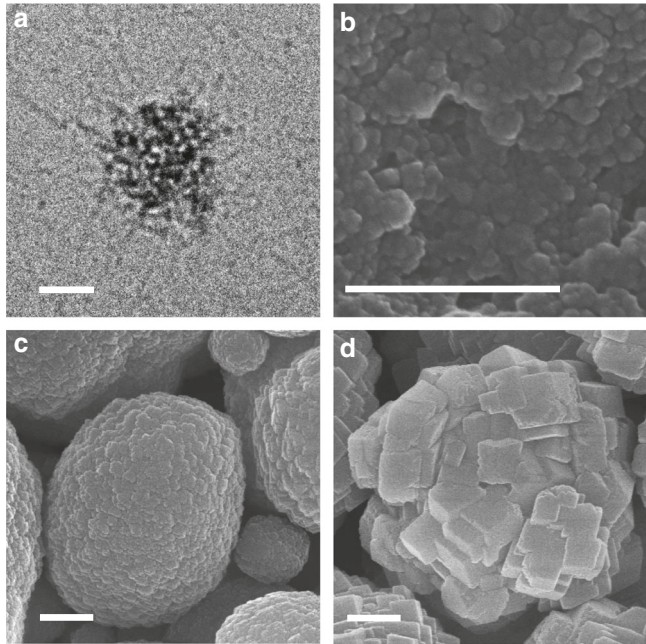

**Fig. 3** Morphological evolution of mineral phases with 2.5 g L$^{-1}$ ds-DNA, 10 mM CaCl$_2$, and slow stirring (100 r.p.m.). **a** CryoTEM image of a NP in the reaction solution at 2 h. **b–d** SEM images of the centrifugal separated 2, 3, and 5 h samples, respectively. Scale bars: **a**, 20 nm, **b–d** 500 nm

with 10 mM of Ca$^{2+}$, the peak shifted −0.13 p.p.m. to high field and broadened (to 0.88 p.p.m.), which is attributed to the interaction between Ca$^{2+}$ and the phosphate groups of ds-DNA. The zeta potential of the neat 2.5 g L$^{-1}$ ds-DNA solution was ~−50.2 mV while the addition of Ca$^{2+}$ let the value to increase to ~−12.6 mV (Fig. 4c, see also Supplementary Table 5), confirming the complexation between Ca$^{2+}$ and ds-DNA. After mixing the solution (10 mM Ca$^{2+}$, 2.5 g L$^{-1}$ ds-DNA) showed a pH of 5.3 and free [Ca$^{2+}$] of 7.25 mM (Fig. 4d, see also Supplementary Table 6), indicating ~2.75 mM of Ca$^{2+}$ was bound to ds-DNA.

Upon the in-diffusion of NH$_3$/CO$_2$ vapor, the pH increased to a stable value of ~10.0 within 1 h (Fig. 4d). Up to a reaction time of 2 h, the free [Ca$^{2+}$] continuously decreased, reflecting the continued binding of Ca$^{2+}$ with CO$_3^{2-}$. At the same time, the liquid $^{31}$P NMR signal shifted slightly back to low field (by 0.03 p.p.m.) and sharpened to a width of 0.72 p.p.m. (Fig. 4a, b), indicating that Ca$^{2+}$ was being released from the binding with ds-DNA due to the formation of CaCO$_3$. During this process also, the zeta potential decreased to −24.2 mV due to the partial release of ds-DNA and/or binding between Ca$^{2+}$ and CO$_3^{2-}$ (Fig. 4c). The increase of pH value was found to have no significant effect to the free [Ca$^{2+}$] and liquid $^{31}$P NMR measurements, while slight decrease of zeta potential was detected due to deprotonation (see Supplementary Fig. 15a, b, Supplementary Tables 5 and 6). This suggests that the interaction between ds-DNA and Ca$^{2+}$ is relatively strong and not affected by the deprotonation of ds-DNA.

A change in slope was observed in both pH and [Ca$^{2+}$] curves at ~120 min, reflecting a change in the soluble [Ca$^{2+}$] and thereby indicating a nucleation event (Fig. 4d). The existence of a nucleation event was confirmed by DLS, which at this point showed an increase in both count rate and average size (Fig. 4c), while cryoTEM showed the appearance of irregular shaped NPs with diameters of ~40 nm (Fig. 3a). FTIR (Fig. 5a) indicated that these NPs are composed of ACC and ds-DNA. The ACC phase is identified by the splitting asymmetric CO$_3$ $v_3$ band at 1409 and

1479 cm$^{-1}$, and the absence of the symmetric $v_4$ vibration at 712 cm$^{-1}$ [49]. The symmetric PO$_2^-$ stretching peak of ds-DNA backbone at 1083 cm$^{-1}$ was more intense in the 2 h sample comparing with the ds-DNA powder, reflecting an interaction between the phosphate group of ds-DNA and ACC[50]. Significantly, SEM showed that these NPs did not coalesce upon centrifugal separation from the solution as would be expected for liquid droplets (Fig. 3b).

The conformation of ds-DNA is also reflected by the $^{31}$P chemical shifts in solid-state NMR (SS-NMR, Fig. 5b, c)[48,51]. The $^{31}$P cross-polarization magic angle spinning (CP-MAS) SS-NMR spectrum of the dry neat ds-DNA showed an asymmetric line shape[52], which can be deconvoluted into two components at ~−2 and −5 p.p.m. (peak 1 and 2). The major component (~80%) at −2 p.p.m. corresponds to the isotropic chemical shift reported earlier for salmon ds-DNA[43]. This line is assigned to ds-DNA in the A-conformation due to dehydration, whereas the resonance around −5 p.p.m. is assigned to B-DNA, which we attribute to the presence of Mg$^{2+}$ ions inhibiting the B-to-A form transition[53]. Ion exchange with Ca$^{2+}$ (see Methods: preparation of Ca$^{2+}$/ds-DNA complex) increases the relative intensity for peak 2 (−5 p.p.m.), indicating a similar interaction of the Ca$^{2+}$ ions with the ds-DNA. A very similar spectrum was obtained for the 2 h sample, in line with the interaction of ds-DNA with Ca$^{2+}$ at this stage of the reaction, and the concomitant stabilization of the B-form of DNA.

Also the comparison of the $^{13}$C CP-MAS SS-NMR spectra of ds-DNA and the 2 h sample (Supplementary Fig. 16) showed the intimate interaction of the CaCO$_3$ with ds-DNA. In the ds-DNA spectrum, the resonance of the base pair carbonyl groups was observed at ~166 ppm[54], while for the 2 h sample this carbonate resonance had a clear shoulder at ~168 p.p.m.[55] Importantly, by performing the $^{13}$C CP-MAS at lower temperature (−100 °C) the relative intensity of the carbonate peak doubled, indicating that at room temperature the ds-DNA/ACC NPs display (anisotropic) dynamics, even in the solid-state.

At ~150 min, another change in the slope of the pH and [Ca$^{2+}$] curves was observed (Fig. 4d), after which the solution became opaque. At this point, DLS showed a further increase in the count rate and average particle size (Fig. 4c), with a maximum at 180 min, together pointing to a second nucleation event. Indeed, after 3 h, micron-sized particles were observed (Fig. 3c) that after 5 h displayed rhombohedral crystal facets (Fig. 3d) and which by FTIR were identified as calcite (Fig. 5a). ICP-OES measurements indicated that the phosphorous mass fraction of the minerals reduced from 18.57 mg g$^{-1}$ for the 2 h sample to 4.52 and 3.77 mg g$^{-1}$ in the 3 and 5 h samples, respectively (Supplementary Table 2), indicating that most of the ds-DNA entrapped in the ACC is released during the transformation to the crystalline state[56], and that the remaining ds-DNA becomes occluded in the calcite crystal. A similar change of phosphorous ratio was also detected by energy dispersive spectroscope (EDS) measurements (Supplementary Table 3).

During crystallization, a continued decrease of the zeta potential to −34.3 mV was observed, in agreement with the release of the ds-DNA/formation of CaCO$_3$ during the process (Fig. 4c). This was confirmed by liquid-state NMR, which showed a further down field shift by 0.08 p.p.m. and a significant sharpening from 0.70 to 0.60 p.p.m. after 160 min (Fig. 4a, b), which is attributed to release of the majority of the ds-DNA during crystallization. After 200 min, the peak showed a width similar to the neat ds-DNA solution, but with a lower chemical shift due to the increased ionic strength related to the introduction of NH$_4^+$ and CO$_3^{2-}$ ions in the solution (Supplementary Fig. 15c, d). In the $^{31}$P CP-MAS SS-NMR spectra of the 3 and 5 h samples (Fig. 5b, c), the fraction of ds-

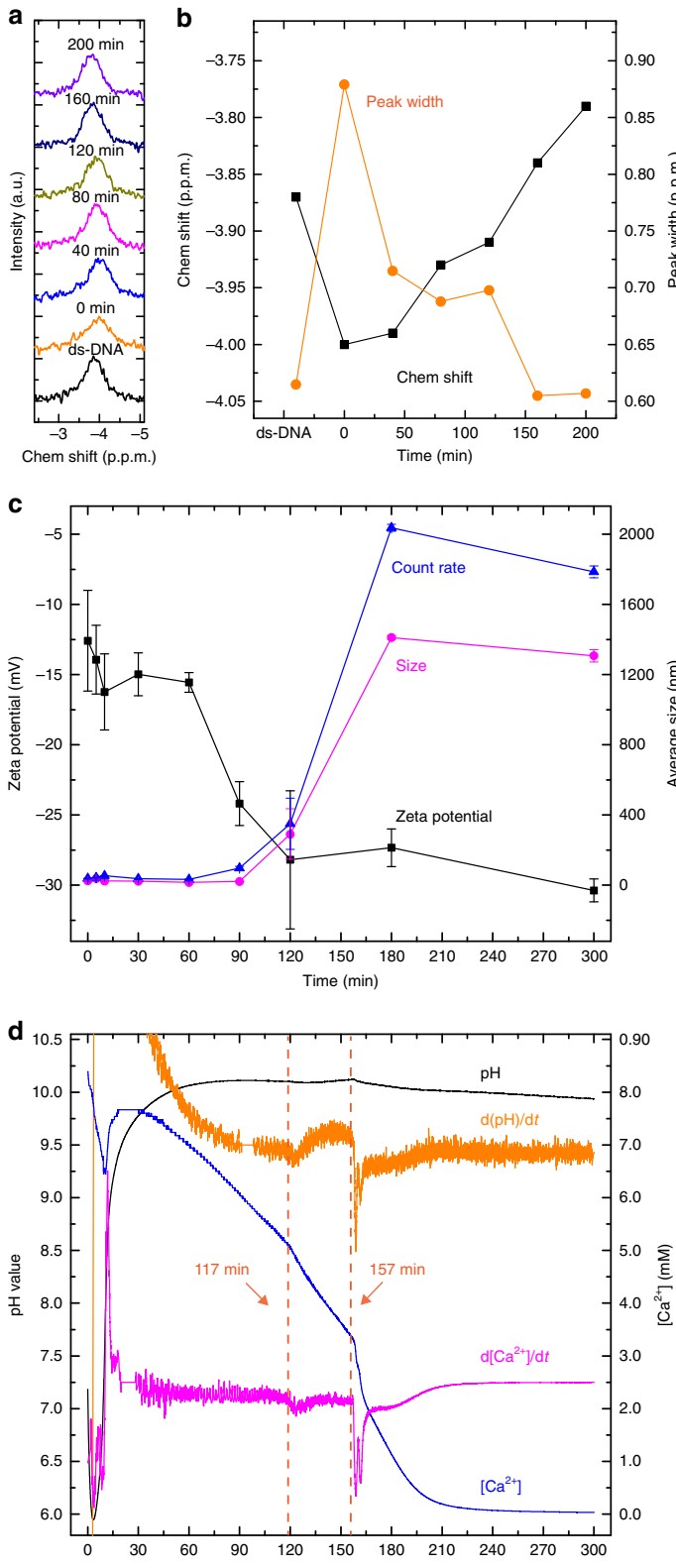

**Fig. 4** Physico-chemical evolution of reaction solutions with 2.5 g L$^{-1}$ ds-DNA, 10 mM CaCl$_2$, and slow stirring (100 r.p.m.). **a** Liquid-state $^{31}$P NMR spectra of a neat 2.5 g L$^{-1}$ ds-DNA solution, and reaction solutions at different growth stages. **b** Chemical shifts and peak widths of the spectra in **a**, which are labeled by black squares and orange circles connected by lines, respectively. **c** Volume averaged particle sizes and count rates of reaction solutions at different growth stages derived from DLS measurements, and corresponding zeta potentials, which are labeled by magenta circles, blue triangles, and black squares connected by lines, respectively. The data were obtained by averaging 10 measurements of each sample, and the error bars represent the standard deviations. **d** pH/[Ca$^{2+}$] curves of the reaction solution and corresponding d(pH)/dt derivative curves. The two changes in slope in the pH and [Ca$^{2+}$] curves at 117 and 157 min are highlighted by orange-dashed lines, respectively

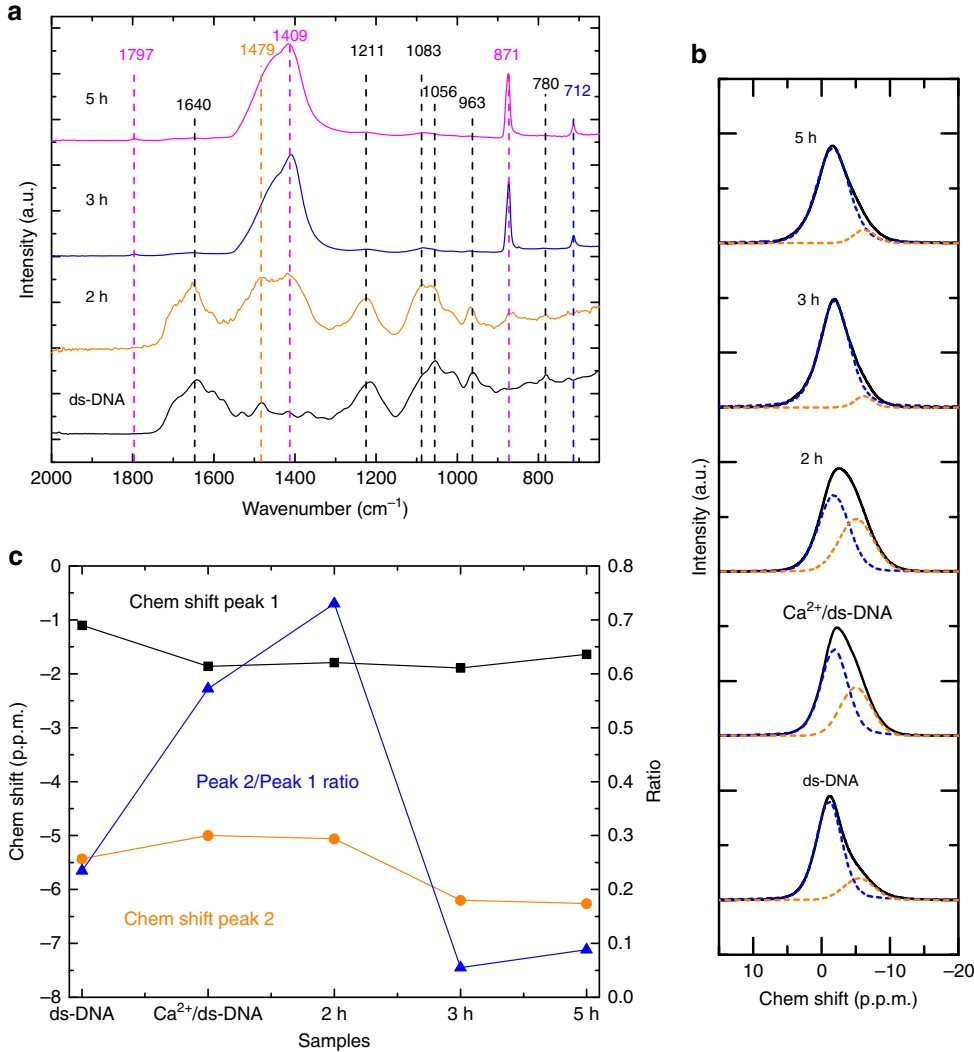

**Fig. 5** Compositional evolution of mineral phases with 2.5 g L$^{-1}$ ds-DNA, 10 mM CaCl$_2$ and slow stirring (100 r.p.m.). **a** ATR-FTIR spectra of ds-DNA and the 2, 3, 5 h samples, respectively. The peaks corresponding to ds-DNA, ACC or calcite are labeled by black, orange and blue-dashed lines and numbers, respectively. The peaks that are shared by ACC and calcite are labeled by magenta-dashed lines and numbers. **b** $^{31}$P CP-MAS SS-NMR spectra of Ca$^{2+}$/ds-DNA complex, and the 2, 3, 5 h samples, and ds-DNA. The signals (black lines) are deconvoluted into two peaks at ~−2 p.p.m. (peak 1) and at ~−5 p.p.m. (peak 2), which are indicated by blue and orange-dashed lines, respectively. **c** Chemical shifts of peak 1 and 2, and peak 2/peak 1 ratios of the spectra in **b**, which are labeled by black squares, orange circles, and blue triangles connected by lines, respectively

DNA interacting with Ca$^{2+}$ (peak 2) significantly decreased, confirming the—at least partial—release of the biopolymer during the crystallization process[56]. It is noteworthy that in contrast to the SS-NMR measurements, only a single $^{31}$P population was identified in all of the liquid-state NMR experiments. Apparently, the ds-DNA remained in the B-conformation under these conditions, and the variation in the structure with changing conditions was only visible through a shift of the resonance and a varying line width. We found no evidence of liquid–liquid phase separation in this system, in contrast to a previous $^{13}$C liquid-NMR study of the pAsp/CaCO$_3$ system[32].

## Discussion
Our experiments reveal that in a solution containing charged polymers such as pAsp, ds-DNA, pAA, or pAH, CaCO$_3$ forms a nanoparticulate phase that has all the solidification and deposition characteristics of the wide spread PILP. Despite its name, this "PILP" phase does not show the microscopic characteristics of a liquid—such as the coalescence of droplets or an extra signal in

liquid $^{31}$P NMR. In fact, PILP consists of ACC NPs with a ~2 nm nanoparticulate texture, and no other product was detected before the formation of this phase in our experiments (Supplementary Figs. 8, 9). The same ~2 nm nanoparticulate texture was found in all the four PILP systems we studied, which strongly suggests that PILP is assembled from ~2 nm clusters. However, these were not detected as individual clusters in the early-stage reaction solutions, which may mean that they are not stable before assembling on the polymers. Our findings could explain several features reported for PILP better than the current droplet-based model, including its gel-like elasticity[33], and the granular morphology of the PILP induced CaCO$_3$ thin films (Fig. 1a)[27,35]. Furthermore, since PILP films deposit to growth substrates by the attachment of NPs rather than wetting of droplets, it is not surprising that only extreme "contact angles" were observed during the process:[34] when the NPs are able (or unable) to attach to the substrate to form a thin film, they behave like a liquid with contact angle = 0° (or >150°). Our proposal of that PILP is assembled from ~2 nm-sized ACC clusters also explains why they could be stabilized by a relatively low concentration of polyelectrolytes e.g.,

$25 \, \mathrm{mg \, L^{-1}}$ of pAsp, which contains only ~0.22 mM of $COO^-$ groups. Instead of binding with all the $Ca^{2+}$ ions, the polyelectrolytes only need to bind with the surface $Ca^{2+}$ of the ~2 nm-sized ACC clusters in PILP. Each of these ACC clusters consists of ~20 $Ca^{2+}/CO_3^{2-}$ ion pairs according to a recent computational study[57]. As a result, much more $Ca^{2+}$ could be stabilized by one $COO^-$ group. Indeed, as shown by a previous compositional study of PILP by Dai et al.[58], the $Ca^{2+}/COO^-$ ratio is ~10:1 for PILP induced by $20 \, \mathrm{mg \, L^{-1}}$ of pAsp, suggesting that each ACC cluster is bound with two $COO^-$ groups in pAsp-induced PILP. As a result, $25 \, \mathrm{mg \, L^{-1}}$ of pAsp could stabilize PILP representing ~2.2 mM of $Ca^{2+}$, allowing a higher concentration of PILP formation. Interestingly, a similar $Ca^{2+}/-PO_4^-$ ratio (10:1) was detected for the ds-DNA induced PILP by the ICP-OES measurements (Supplementary Table 2), and it remains unclear why a much higher concentration of ds-DNA ($2.5 \, \mathrm{g \, L^{-1}}$) is required to induce the PILP process.

Apparently, our results contrast with the NMR work of Bewernitz et al.[32] who used a [13]C enriched titration system with high $CO_3^{2-}/HCO_3^-$ concentration, moderate pH (~8.5) and presence of pAsp ($18 \, \mathrm{mg \, L^{-1}}$). An extra [13]C liquid-NMR signal was detected in this system after $Ca^{2+}$ titration, which they attributed to PILP, although it was not shown that typical products of PILP processes (thin film or nanorods) could form in such a system. The signal showed a $T_2$ (spin–spin) relaxation time slightly shorter than the one of the solution but orders higher than regularly encountered in solids. Although the authors used these results to support the liquidity of PILP, the long $T_2$ of PILP could also be attributed to the fast dynamics of very small ACC clusters as observed in our experiments. In fact, the self-diffusion measurements of Bewernitz et al. showed an effective size of ~1.6 nm of PILP, which agrees very well with the size of ACC clusters we observed (~2 nm). Hence, our results are consistent with previous reports in spite of the differences in experimental conditions, and all data can be interpreted such that PILP is a dispersion of dynamic assemblies of ~2 nm-sized ACC clusters that are cross-linked by charged polymers, without necessarily the presence of a liquid–liquid phase separation.

Despite its granular character, the PILP phase behaves like a liquid on a macroscopic/mesoscopic scale, being able to "wet" many solids surfaces and to infiltrate pores, which is quite different from solid ACC. Solid ACC NPs have a continuous structure, and usually dissolve within several minutes in aqueous solution[45]. In contrast, the polymer-ACC NPs possess a bicontinuous internal structure composed of ~2 nm-sized ACC clusters that can exist in solution for up to several hours. The observed structure suggests that the polymers are intertwined with and bound to the ACC clusters, stabilizing them inside the NPs, as illustrated in Fig. 6a (stage 2). This temporarily inhibits the ACC-calcite phase transformation, whereas the surface bound polymers colloidally stabilize the NPs, allowing them to behave macroscopically like a liquid phase. Similar liquid-like behavior has been reported for surfactant functionalized colloidal particles[59,60]. Due to steric repulsion of neighboring functional groups, these colloidal particles could flow even after drying[60]. The polymers are excluded during crystallization to minimize the free energy and stress of the bulk crystal[56], and the dynamics of the polymer-mineral interactions—as indicated by [13]C CP-MAS (Supplementary Fig. 16)—will play a vital role in the process. The PILP NPs will completely cover surfaces that match with their ACC/polymer surfaces (or not at all when the surfaces are incompatible with the surface of PILP)[34]. This enhanced "wetting" behavior facilitates the nucleation of shapes with non-conventional morphologies within nanopores or on flat substrates, which only afterwards transforms into crystalline $CaCO_3$ nanorods or films. The concave tips of the nanorods grown in nanopores

(inset of Fig. 1d) are also explained by the affinity of the polymer modified clusters to the pore wall (Fig. 6b, with polymer), as an alternative to the previously proposed wetting by a fluid-like PILP phase[21]. In the absence of such polymers (Fig. 6b, without polymer), the ACC-calcite transition is much faster under the experimental conditions used here (Supplementary Fig. 3), and it is much more difficult for these crystals to grow into nanopores, or to "wet" a solid surface where they must adapt their shape to non-equilibrium morphologies with high surface energy (Supplementary Fig. 5), which will limit their growth. It is noteworthy that the nanorods we described are different from the $CaCO_3$ nanorods formed in the bulk solution, as has been reported for pAA and pAH induced PILP processes[19,61]. Those nanorods are however not observed in our experiments, even when pAA or pAH was used, and their detailed formation mechanism so far remains unresolved.

Based on our results, a mechanism for the polymer-induced formation of $CaCO_3$ thin films and nanorods is proposed in Fig. 6. Taking negatively charged polymers as an example (Fig. 6a), the reaction starts with the formation of a $Ca^{2+}/polymer^-$ complex (stage 1, see also Supplementary Tables 5 and 6). Upon reaction of $CO_3^{2-}$ with the polymer-bound $Ca^{2+}$ ions, ~2 nm-sized ACC clusters develop, which are cross-linked and stabilized by polymer molecules, forming 30–50 nm-sized composite polymer-ACC NPs (stage 2, see also Fig. 2a). The NPs aggregate into micron-sized structures in bulk solution (stage 3, see also Fig. 1e). In the presence of a flat substrate, the NPs deposit to form thin films (stage 4, see also Fig. 1a). Nucleation of crystallized $CaCO_3$ happens later, generating crystalline plates within the thin film (stage 5, see also Fig. 1f). Since the ACC phase is hydrated (usually ~1:1 $CaCO_3$: $H_2O$)[62], and the polymers within PILP will be excluded after crystallization, a larger volume of PILP is consumed to form the same molar amount of dehydrated crystalline polymorphs such as calcite or vaterite. This then lead to a dissolution of the ACC thin film around the growing crystalline platelets, which is promoted by the decreasing supersaturation of the solution (Fig. 4d) due to the relatively higher solubility of ACC comparing with calcite or vaterite. After complete crystallization (stage 6, see also Fig. 1b), the majority of polymers are excluded from the bulk crystals to the crystal surfaces or back to the solution, while some of them become occluded. When a track etch membrane is exposed to the reaction solution (Fig. 6b), short calcite nanorods with crystal facets on their tips are formed in the absence of polymers (see also Supplementary Fig. 5). In the presence of negatively charged polymers, however, ~2 nm-sized ACC clusters covered by polymers are formed, which efficiently deposit on the walls of the nanopores and form nanorods with concave tips (Fig. 1d).

Our results provide new insights into the mechanism of this wide spread crystallization strategy, showing that the thin film formation and infiltration of nanopores observed from PILP systems can be explained by the attachment of polyelectrolyte-stabilized colloidal particles to surfaces[63]. Also in biological systems similar biopolymer-stabilized nanoparticulate assemblies may form the precursors for mineral formation[36–38]. Indeed the relatively high solid content (~50 volume %) of the NPs can form an efficient source of mineral and may constitute the mineral content of the vesicles observed in active mineralizing systems[38,64]. The involvement of such a biopolymer-mineral based PILP system as precursor in biomineralization would explain the nanogranular textures observed in many systems[36–38,64], as well as the infiltration of preformed macromolecular templates such as collagen in bone and the organic matrix of the nacre in mollusk shells.

In conclusion, we show here that the $CaCO_3$ PILP consists of gel-like NPs, which should be formed by the assembly of

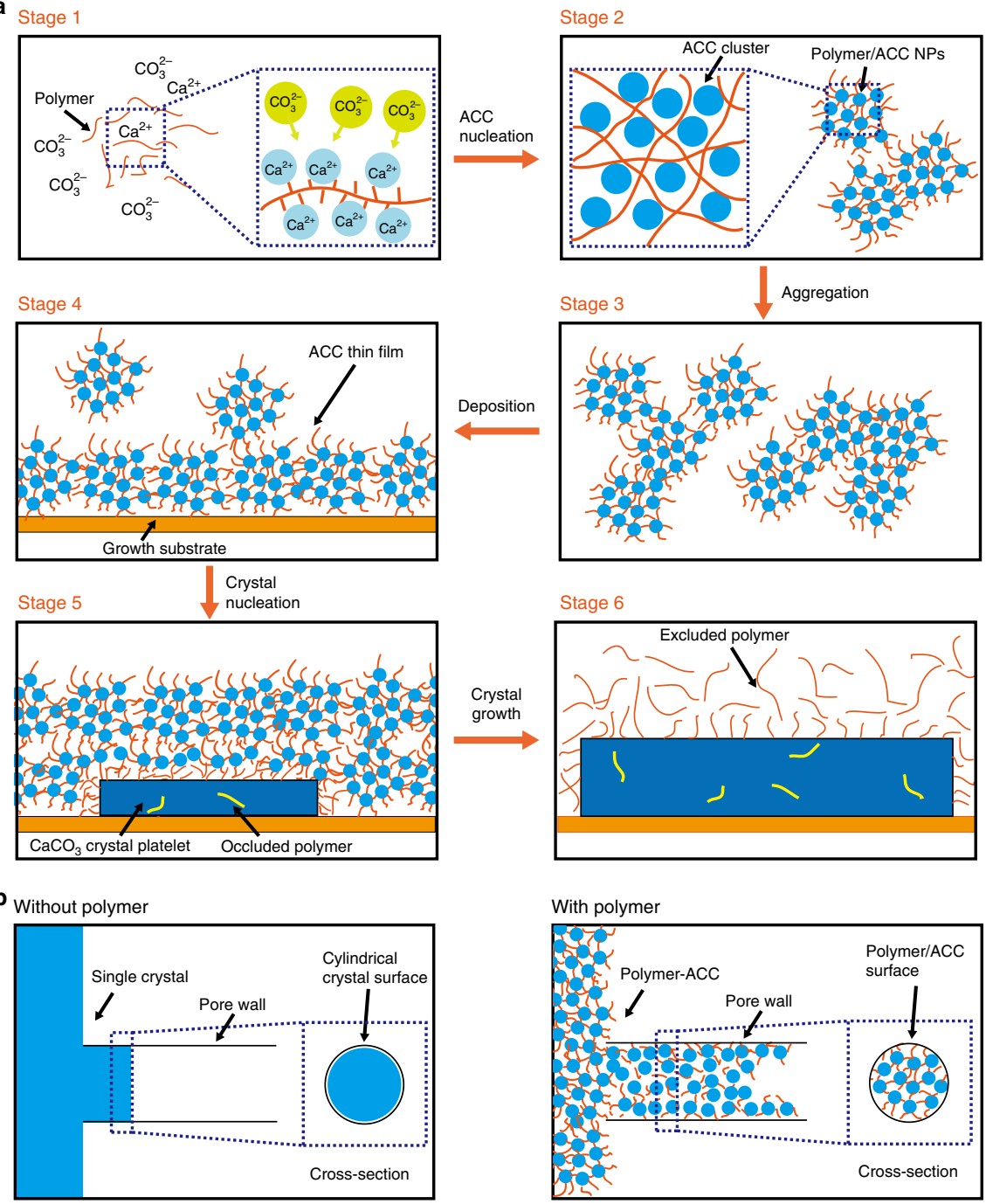

**Fig. 6** Scheme of the CaCO$_3$ thin film or nanorod formation processes induced by negatively charged polymer. **a** CaCO$_3$ thin film formation process starts with the formation of a Ca$^{2+}$/polymer complex (stage 1). Upon binding of CO$_3^{2-}$, ~2 nm-sized ACC clusters develop from the polymer-bound Ca$^{2+}$, which are further cross-linked and stabilized by polymer molecules, forming 30–50 nm-sized polymer-ACC NPs (stage 2). The NPs aggregate into μm-sized structures in bulk solution (stage 3). In the presence of a flat substrate, the NPs deposit to form thin films (stage 4). Subsequently nucleation of CaCO$_3$ crystals (stage 5) generates platelets within the thin film. The majority of polymer molecules is excluded after crystallization (stage 6), while some of them become occluded inside the crystal. **b** CaCO$_3$ nanorods are formed when a track etch membrane is exposed to the reaction solution. Short calcite nanorods with crystal facets on their tips are formed without the presence of polymers. In the presence of negatively charged polymers, the ~2 nm-sized ACC clusters covered by polymers deposit to the walls of the nanopores and form nanorods with concave tips

polymer stabilized ~2 nm ACC clusters. Although a transient dense liquid phase has been demonstrated for CaCO$_3$[29,57,65,66], our results show that CaCO$_3$ PILP is not simply a polymer stabilized version of this liquid phase. More specifically, the assemblies of ~2 nm clusters show no sign of nanoscopic fluidity, and do not coalesce into continuous structures in

solution. We therefore suggest to use the abbreviation PILP for polymer-induced liquid-like precursor, acknowledging the exceptional macroscopic properties of these polymer directed crystallization systems. The presented model for PILP not only aids to our understanding on how control over crystal morphology can be achieved in materials of technological relevance,

it may also provide mechanistic insights into the formation processes of minerals in biological systems.

## Methods

**CaCO₃ mineralization reaction.** Analytical grade $CaCl_2$, $(NH_4)_2CO_3$, poly-(α, β)-DL-aspartic acid sodium salt (pAsp, mw = 2000–11000 g mol⁻¹), poly (acrylic acid) (pAA, mw = 5100 g mol⁻¹) and poly(allylamine hydrochloride) (pAH, mw = 15,000 g mol⁻¹) were purchased from Aldrich. Salmon sperm double-stranded DNA (300 bps, mW ≈ 225,000 g mol⁻¹) was provided by Maruha Nichiro Co. Ltd., Japan. The mineralization of $CaCO_3$ is induced by diffusion of $CO_2/NH_3$ mixture gas from $(NH_4)_2CO_3$ powder into $CaCl_2$ solutions:[18,67] A 80 mL beaker containing 25 mL solution with 10 mM $CaCl_2$ and 2.5 g L⁻¹ ds-DNA[23], 25 mg L⁻¹ pAsp[18], 25 mg L⁻¹ pAA[24], 25 mg, or 1 g L⁻¹ pAH[19] was placed in a desiccator (Supplementary Fig. 1a). A hydrophilic glass slide was vertically inserted into the solution as the substrate for $CaCO_3$ growth. The beaker was covered by a holey parafilm to reduce the diffusion speed. A vial containing 1 g of $(NH_4)_2CO_3$ powder was placed near the solution. The $CO_2/NH_3$ mixture gas released by $(NH_4)_2CO_3$ decomposition was slowly diffused into the solutions to induce the $CaCO_3$ formation. The growth was allowed to proceed over periods from 10 min to 24 h, with no stirring or 100 r.p.m. stirring. Reaction solutions were collected near the solution/air interface for cryoTEM visualization. Samples grown on the glass slide were thoroughly washed using ethanol and dried at room temperature for 24 h before characterization. For experiments with stirring (Supplementary Fig. 1b), the pH electrode and $Ca^{2+}$-ion selective electrode connected to a Metrohm Tiamo system were directly inserted into the reaction solution for measurements. The as-prepared samples were centrifuged for once in the reaction solution, then centrifuged for two more rounds in ethanol, then dried at room temperature for 24 h before characterizations. The average relative centrifugal field is 4715×g for all of the centrifugations in our experiments.

**CaCO₃ nanorod formation within track etch membrane.** $CaCO_3$ nanorods are formed within track etch membranes with nano-sized pores also using $(NH_4)_2CO_3$ diffusion method[20,21]. A 10 μm thick poly-carbon track etch membranes with 50 nm (item no. 515-2026, supplier no. 110603) or 200 nm (item no. 515-2029, supplier no. 110609) sized pores were ordered from VWR. The membranes were plasma cleaned for 1 min before using to improve the hydrophilicity, then immersed into 10 mM $CaCl_2$ solutions with 2.5 g L⁻¹ ds-DNA or 25 mg L⁻¹ pAsp or no additive. The solutions were vacuum degassed overnight to remove the gas left within the pores. After that, $CaCO_3$ was grown by the $(NH_4)_2CO_3$ diffusion method as mentioned above for 24 h. The membranes were then thoroughly washed by ethanol, and wiped by filter paper to remove the crystals formed on the membrane surface. Then, the membranes were dissolved in dichloromethane ($CH_2Cl_2$) and ultrasonically treated for 20 min to separate the $CaCO_3$ products. The products were then centrifuged for three cycles in $CH_2Cl_2$ and two cycles in ethanol, and dried at room temperature for 24 h before characterizations.

**Preparation of Ca²⁺/ds-DNA complex.** $Ca^{2+}$/ds-DNA complex was prepared by mixing 60 mL of solution containing 10 g L⁻¹ ds-DNA and 300 mM $CaCl_2$ (prepared using deionized water pre-cooled to 4 °C) with 140 mL of pre-cooled (−20 °C) ethanol. The precipitates were centrifuge washed using pre-cooled (−20 °C) 70% ethanol for five rounds to remove the remaining $CaCl_2$ salt. The as-prepared complex was dried at room temperature before characterization. EDS shows the Ca:P:Cl ratio in the complex is ~1:2.28:0.14 (Supplementary Table 3), indicating the $Ca^{2+}$ was bound to the ds-DNA while most of the $CaCl_2$ salt was removed.

**SEM/EDS and OM observations.** SEM and EDS studies were performed using a FEI Quanta 3D field emission SEM equipped with an EDAX EDS detector. In situ optical microscope observation of $CaCO_3$ thin film formation process was performed in a specifically designed growth chamber (Supplementary Fig. 6). In the growth chamber, 0.12 g of $(NH_4)_2CO_3$ was loaded in a lower cell B. The $CO_2/NH_3$ mixture gas released by $(NH_4)_2CO_3$ decomposition was slowly diffused into a higher cell A containing 39 mL of reaction solution with 10 mM $CaCl_2$ and 25 mg L⁻¹ of pAsp. The products were formed on the cover glass and observed using a Zeiss AxionVision2 optical microscope. The video of in situ observation was recorded with a rate of 1 frame per 3 s in transmission mode with a ×32 objective lens and DIC filter.

**Cryogenic transmission electron microscopy.** Three microliters of reaction solution samples were applied on a quanti-foil or lacey TEM grid, which was blotted for 5 s and vitrified using an automated vitrification robot (FEI Vitrobot™ Mark III, FEI Company). For the tomography experiments, the liquid samples were mixed with 5 nm-sized gold markers before blotting. To study the early-stage samples, 3 μL of reaction solution samples were applied on a GOx-coated TEM grid[41], and 20% (v/v) IPA in ultrapure water was used in humidifier. After 60 s of waiting, the grid was blotted for 3 s and vitrified. CryoTEM imaging was performed under ~3 μm defocus on a FEI-Titan TEM equipped with a field emission gun and operating at 300 kV. For samples on GOx-coated grids, imaging was done using a parallel beam with an illuminated area of 670 nm (nanoprobe), with a defocus

value of −1.5 μm. Images were recorded using a 2k × 2k Gatan CCD camera equipped with a post-column Gatan energy filter (GIF), with an electron dose of 15.8 e Å⁻² per image. The tomography tilt series were taken from −65 to 65°, 3° per step, with an electron dose of 3.0 e Å⁻² per frame. For the discussions on the limitations of cryoTEM, please see refs. [30,40].

**Morphological processing and analysis of tomographic results.** The tomographic tilt series were aligned and reconstructed in IMOD Etomo. The final aligned stack was binned by 2 and reconstructed using the SIRT algorithm with 10 iterations. The gold markers were filtered from the final reconstruction by setting the intensity of the gold beads in the tilt series to the average background intensity. For the tomographic results shown in Fig. 2g–k and Supplementary Movie 3, the reconstruction stacks were median-filtered (filter size 3 × 3 × 3 pixels) to remove noise for better visibility. The comparison between original stack and filtered stack is shown in Supplementary Fig. 17.

Supplementary Movie 2 was generated from the tomographic results of the NP shown in Fig. 2g–j, with a further filtering process: different filtering levels were applied to the NP and the remaining volume of the reconstruction using in-house Matlab scripts, to segment the mineralized part of NP, enhance the contrast for ds-DNA, and then show both together:

1. The reconstruction (stack 0) was 3D median-filtered (filter size 3 × 3 × 3 pixels) to generate stack 1, in order to remove noise for better visibility.
2. All the objects in stack 1 were segmented by using the intensity threshold obtained by Otsu's method (−384), generating stack 2.
3. The segmented objects were eroded by a diameter of 2 pixels to remove noise artifacts and pick up the A area, generating stack 3.
4. The A area in stack 3 was dilated by a diameter of 10 pixels to create a continuous mask (B area). B area is used as a mask to the stack 1.
5. B area in stack 1 was segmented by the Otsu's method-derived intensity threshold to get the low pixel intensity part of NP (C area), which corresponds to the mineralized part of the NP that has higher density and shows lower pixel intensity.
6. Stack 0 was 3D median-filtered (filter size 7 × 7 × 7 pixels) to increase the contrast of ds-DNA, generating stack 4.
7. C area was applied as a mask to stack 4 and set to black, i.e., showing the accurately segmented ACC particles as black, generating the final stack (stack 5), which is used for generating Supplementary Movie 2. A comparison between the as-filtered stack and the original stack is shown in Supplementary Fig. 18a and b, which shows that the contrast of ds-DNA is significantly improved, while the morphology of NP is preserved.

The C area (mineralized part of the NP) was used to measure the volume ratio of the mineralized part and its connectivity. It was found that the >99% of the NP object area is interconnected. B area was eroded by a diameter of 10 pixels to create an area that tightly encloses the NP (B′ area), C area was subtract from B′ area to generate the D Area, which corresponds to the ds-DNA/water-filled area that has lower density and shows higher pixel intensity. This D area is also interconnected together, indicating the NP possesses a bicontinuous structure. The volume proportion of the mineralized part (C area: B′ area) is ~51%. The reliability of the value was confirmed by using different pixel intensity thresholds and comparing with the results obtained from stack without filtering (Supplementary Table 4).

**Nuclear magnetic resonance measurements.** Liquid-state ³¹P NMR spectra of reaction solutions at different stages of the reaction were obtained on a Bruker Avance 500 MHz spectrometer equipped with a 5 mm Bruker PABBI broadband inverse probe. The reaction solutions were mixed with 10% $D_2O$ before measurements to provide a lock signal. Spectra were recorded at 25° accumulating 256 scans with recycle delay of 5 s, and referenced with respect to trimethyl phosphate (TMP).

Solid-state ³¹P and ¹³C NMR spectra were recorded on a 9.4 T solid-state Varian NMR system (VNMRS) using a Varian 3.2 mm T3-HXY MAS probe, configured in double-resonance mode for ¹H-³¹P and ¹H-¹³C, respectively. The samples (except ds-DNA powder, which was measured as it is) were isolated by centrifugation, dried at room temperature, and directly packed into the rotor without further treatment. The ³¹P spectra were recorded at a spinning speed of 16 kHz, a spectral width of 100 kHz, and an acquisition time of 20 ms. ¹H decoupling was applied in all cases using the SPINAL sequence[68] at an rf-field strength of 70 kHz. Spectra are referenced with respect to 85% phosphoric acid. For the direct excitation measurements a recycle delay of 500 s was used. The cross-polarization experiments were recorded using a recycle delay of 10 s and a CP contact time of 0.8 ms. The DE ³¹P spectra of all the samples are indistinguishable from their CP spectra due to the abundant protonation of ds-DNA. ¹³C CP-MAS spectra were recorded at a spinning speed of 16 kHz, a spectral width of 100 kHz, and an acquisition time of 40 ms. SPINAL ¹H decoupling was applied employing an rf-field strength of 100 kHz. The CP contact time was 4 ms and the recycle delay was 3 s. Spectra are referenced with respect to TMS using adamantane as a secondary reference.

**FTIR/Raman and zeta potential/DLS measurements.** ATR-FTIR spectra of the ds-DNA powder or centrifuge-separated and room temperature-dried precipitates

were obtained directly on the samples using a Varian FT-IR 3100 Spectrometer with Golden Gate ATR accessory and were signal-averaged over 50 scans at a resolution of $4\,cm^{-1}$. Raman measurements of the samples grown on glass substrates were performed using a Jobin Yvon LABRAM confocal-Raman spectrometer (Horiba). An Olympus BX40 optical microscope with a ×100 objective lens was used to find the product of interest, and a laser of wavelength of 633 nm was focused on sample using a D3 filter with a hole size of 200 μm. An acquisition time of one minute was used to obtain the spectra with a resolution of $4\,cm^{-1}$ in the range of $100–2000\,cm^{-1}$. The zeta potential/DLS measurements were performed on 1 mL of reaction solution collected at different experimental stages using Malvern Instruments Zetasizer (Nano ZS) with a 633 nm laser.

**ICP-OES measurements.** ICP-OES with end on plasma (Axial plasma) were performed using a SPECTROBLUE EOP spectrometer (AMETEK, Germany), with a measurement wavelength range of 165–770 nm. The generator of the spectrometer runs at 27.12 MHz, 1.4 kW. The samples were dissolved in 5 mass% HCl and heated at 120 °C in thick wall Teflon bottles for 16 h before the measurements according to a previous report[69], in order to fully digest the ds-DNA molecules in the samples for a more accurate phosphorous measurement. The bottles were weighted before and after heating to measure the mass loss due to evaporations (<0.2%). Different dissolution concentrations (from 4 to $80\,mg\,L^{-1}$) were used for the samples in order to make sure that the signals of different elements all fall within the detection limits. The solutions were introduced to the spectrometer by a cross-flow nebulizer and Scott spray chamber, with a sample uptake rate of $2\,mL\,min^{-1}$ and argon gas flow. The calibration spike solutions for Ca, P, Mg, and Na were prepared from the standard ICP solutions (VWR), respectively. The possible disturbance of ds-DNA to the plasma was tested for solutions with different ds-DNA concentrations, and no influence was detected for up to $1\,g\,L^{-1}$ of ds-DNA.

**Code availability.** All the Matlab scripts used for the morphological processing and analysis of tomographic results are available online at: Raw Data for "Microscopic structure of the polymer-induced liquid precursor for calcium carbonate," Figshare, https://doi.org/10.6084/m9.figshare.6340547.

**Data availability.** All the data that support the findings of this study are available online at: Raw Data for "Microscopic structure of the polymer-induced liquid precursor for calcium carbonate," Figshare, https://doi.org/10.6084/m9.figshare.6340547.

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

## Acknowledgements
The work of Y.X. was financially supported by the China Scholarship Council (CSC) and by a TOPPUNT from the Netherlands Organization of Scientific Research (NWO) to N.A.J.M.S. NWO is also acknowledged for their support of the solid-state NMR facility for advanced materials research. A.A. is an Awardee of the Weizmann Institute of Science-National Postdoctoral Award Program for Advancing Women in Science. We thank Jan F. Schoonbrood and Ruud Aspers (Radboud University Nijmegen, The Netherlands) for help with NMR measurements, Adelheid Elemans-Mehring and Emiel J.M. Hensen (Eindhoven University of Technology, The Netherlands) for help with ICP-OES measurements, Pauline Schmit (Eindhoven University of Technology, The Netherlands) for help with OM measurements, Hanglong Wu and Arthur D.A. Keizer (Eindhoven University of Technology, The Netherlands) for discussions of the morphological analysis, Jozua Laven (Eindhoven University of Technology, The Netherlands), Anna B. Spoelstra (Eindhoven University of Technology, The Netherlands) and Wouter J.E.M. Habraken (Max Planck Institute of Colloids and Interfaces, Germany) for beneficial discussions, and Maruha Nichiro Co. Ltd., Japan for kindly providing the ds-DNA molecules.

## Author contributions
Y.X. carried out most of the experiments and co-wrote the manuscript. K.C.H.T. carried out the SS NMR experiments. P.H.H.B. provided support with the cryoTEM. A.A. provided support with NMR result analysis. H.F. provided support with tomographic reconstructions and morphological analysis. A.P.M.K. supervised the NMR experiments and data analysis. N.A.J.M.S. supervised the project and co-wrote the manuscript. All authors discussed the results and revised the manuscript.

## Additional information

**Competing interests:** The authors declare no competing interests.

