## [Peer Review File · Nature Communications]

Reviewers' comments:

Reviewer #1 (Remarks to the Author):

The manuscript describes amorphous calcium carbonate precursors formed in presence of various polyelectrolytes, positive and negatively charged polyelectrolytes, (Pasp, PAA, DNA and PAH, with main focus on DNA). The authors challenge the concept of widely accepted precursor phase so called "PILP"(polymer induced liquid precursor). The authors argue that this specific phase is not a liquid at nano/microscale, despite of their macroscopic liquid-like behaviors observed by many researchers for many years. The authors convey their argument by using CryoTEM and NMR mainly. The results show the primary clusters formed in presence of the polyelectrolyte in the very early stage are ~2 nm in diameter, which don't coalesce as you would expect from liquid, therefore conclude the acronym PILP is misleading in some extent.

The quality of the manuscript is very good and comprehensive overall, but one of key characterizations, NMR data is only limited to DNA/CaCO₃ case, which is much less commonly accepted for PILP process, compared to ones with Pasp or PAA. It is a convenient choice as it allows to detect P-NMR signals, but in the weak side, this causes rather large discrepancy in concentrations (eg. 100 times more than reaction using other polyelectrolytes, 25mg/ml). More importantly, while the authors intend to compare their NMR data to the results from the previous NMR work from Gower group, I found that not only different polyelectrolytes, (DNA.PILP vs Pasp/PILP) were used, but also solution conditions and precipitation method are entirely different. I am not very convinced by the fact the two systems are equivalent to directly compare. So the main question still remains. In the same line, the authors emphasize generality of their observations, but overlook differences from specific cases. For example, while Pasp and DNA cases appear similar, PAA and PAH don't, by judging the size or structures of aggregates (figure 2 vs figure s8) I looked up the original statement about PILP, and found out in many papers, the phase has already been described as "liquid-like" behavior. Therefore, the closing remark of this paper "we therefore suggest that the abbreviation PILP for polymer-induced liquid-like precursor" would not convey any strong message as the authors wish. I suggest to reconsider it. This is thorough studies with many beautiful data, particularly the cryo-TEM images and videos are striking, I am sure that crystallization/biomineralization community could benefit from. Therefore I would encourage publication if the authors address the comments and questions, hoping it would strengthen the paper.

Specific points

1. Assuming those aggregates (very much uniform sizes) are coming from 2 nm clusters, one would expect to see many individual 2nm clusters (single/double/triple...) in surrounding media. I found this most strange. This makes me think whether the very early stage was missed. From schematic/mechanism of the process, should Ca/polyelectrolyte complexes should be the first products? Followed by formation of many individual 2 nm clusters ?

The authors also mentioned on page16 "PILP consists of assemblies of 2 nm sized ACC clusters that are physically cross-linked by charged polymers, and no other product was detected before the formation of this phase in our experiment"

2. The concentrations of polyelectrolytes are too low to stabilise all the 2 nm clusters. Taking Pasp (Mw 10000) for an example, it is only 1-2 μ M, which is extremely low to stabilize mineral phase (10 mM) used here ?

Moreover as the authors claim, If the wet phenomenon were due to polyelectrolyte stabilizing the clusters, then demonstration of same phenomenon with the same polyelectrolyte stabilized (solid)colloids would be the strongest and direct evidence? PAH or PAA are widely used to stabilize colloidal metal/inorganic nanoparticles in similar sizes, like Au clusters.

Authors could benefit from discussion further regarding "nanofluid" behavior : Generalized Route to Metal Nanoparticles with Liquid Behavior. J. Am. Chem. Soc. 128, 12074–12075 (2006).

3. Line 133 page 7 "did not coalesce to form continuous spherical objects" if I understood correctly, f and l in Figure S8 represent "liquid" nature of objects. However, even those doesn't look very spherical, even though I do agree with the authors the absence of internal subunits.

Out of curiosity, "nanoscopic fluidity" has ever been directly observed by in-situ EM ?

4. In Figure 1. The crystalline platelet is formed by dissolution of ACC films, it contradicts again with the suggested mechanism/schematic here and the original hypothesis of PILP, where solid state transformation is implied ?

5. NMR : the pH is not constant during the reaction, changes in ionic strength and pH in the solution will heavily affect the interaction and conformation of DNA and might diminish or enhance such effects. Therefore it is not clear that how to decouple the effect of interaction to ACC from pH effects, unless data of DNA/Ca at varied pH support otherwise.

Minor comments

Line 62 page 3 "with unprecedented detail the formation and transformation"

I would not agree with this.

Line 100 Page 4 "~extending over centimetres" all figures shown only show micron-meter scale, I believe it would be typo.

Using unified units, hour or mins, will help readers

Supplementary figure s7 : in raman spectra (f), what do the humps around 500-600 and 800 cm^{-1} come from, doesn't match with ACC ?

Supplementary table 5 : measurement at actual reaction pH (-9-10) would be more informative.

Reviewer #2 (Remarks to the Author):

The authors studied the PILP process using advanced analytical techniques such as CryoTEM, Raman, and NMR techniques. The study highlights the mechanism behind mineral formation process, which may be relevant to biomineral formation or synthesis of novel biomimetic materials. The work is important. However, some of the results and discussion may be more convincing after more analysis and review of prior literatures.

Line 85-95: Figure 1 e-h and supplementary video 1 describes the formation of thin calcite crystals at the later stage. However, the reviewer highly doubts the dissolution of ACC film in regions over 200 μm diameters resulted in the formation of the calcite crystal. It is highly possible that ACC film still there and polymer is completely consumed at the late stage, the remaining Ca (which may be released from dissolution ACC phase as well) will react with CO_3 to form Calcite crystals. The observed "dissolution" may be just an artifact of focusing of objective lens on the crystals. If calcite crystal formed at the expense of dissolution of ACC phase, one should observe the disappearance of ACC film close to the crystal first, then extend the disappearance to outside. In summary, the reviewer doubts the observed disappearance of ACC from outside to inside.

Line 246-251: The authors talked about the release of Ca from Ca-DNA complex, which is correct. However, when Ca is released from Ca-DNA, there is already CaCO_3 phase formation. The observed Z-potential change measurement is based on particles. At this stage, there are two types of particles that contribute to the Z-potential: ACC particles and maybe macromolecular DNA particles. Thus the

change of Zeta-potential does not confirm the release of Ds-DNA.

Line 110-111: Crystal growth at air-liquid interface and on glass slides substrates are different matters. Has Gower mentioned that 2 μm sized particles in the air/liquid interface formed the film on the glass slide? The film formation refers only to the CaCO_3 on glass substrates.

Line 317, Figure 6. The reviewer disagree with picture depicted in stage 6. Can the author proves there are free "excluded polymer" in the solution. During mineral formation, the polymer-Ca complex even release Ca to form CaCO_3 , the polymer stills bounds to mineral surface and polymers imbedded in the mineral phase (the polymer-poor region forms crystal, the polymer-rich region remains ACC phase). The close association of polyaspartic acid with CaCO_3 mineral during and after the PILP process has been demonstrated by Gower's group by using the FITC-labeled Polyaspartic acid. Please confirm the existence of free polymer as the author demonstrated in Stage 6. There is a mechanism figure in the below reference which may be cited and discussed for comparison purpose. "Dai, L. J.; Cheng, X. G.; Gower, L. B., Transition Bars during Transformation of an Amorphous Calcium Carbonate Precursor. *Chemistry of Materials* 2008, 20, (22), 6917-6928."

Line 443, EDS used to quantify the Ca:P:Cl ratio was not an accurate method and the quantification is only accurate on smooth surface and. A better method such as dissolving the material and measure by other technique including ICP-AES should be used.

Reviewer #3 (Remarks to the Author):

This paper brings us much closer to understanding the microscopic structure of the PILP phase. Using cryoTEM, liquid and solid-state NMR, as well as DLS, ATR-FTIR, and conventional microscopies, the authors provide a thorough and in-depth characterization of the early stages of PILP, and provide a unifying theory of its structure. This type of characterization has been lacking in the community, and I am grateful to the authors for providing us with such a clear picture of a very useful but poorly understood, until now, method for obtaining calcium carbonate structures with a range of non-equilibrium morphologies. The characterization is performed with attention to detail and the resulting conclusions regarding the formation mechanism are well-reasoned and supported by experimental data. For these reasons, I recommend publication in *Nature Communications* after minor revisions. Below are specific comments for the authors to consider when revising their manuscript.

Specific comments:

1) It would be helpful to have a clear and concise summary of the criteria for liquid vs. nanoparticulate phases for the non-expert reader who hasn't followed the years of PILP debate and discussion. Such a summary could go into the end of the introduction or beginning of the results section. Something along the lines of what characteristics in TEM, NMR, etc. would indicate liquid vs. nanoparticulate composite. The authors scatter these criteria throughout the manuscript, and I think it would be good to put them all into one paragraph early in the manuscript.

2) New data regarding the growth of calcite nanorods into track-etched membranes in the absence of additives is introduced in the Discussion section (Fig. S12). Please introduce all new results in the results section. I also suggest emphasizing that the reason the track-etch membrane results with and without additives are presented is because the ability to infiltrate pores and form non-equilibrium shapes has been attributed to the liquid-like nature of PILP (see comment #1 – if the track etch membrane experiments are motivated as one of the observations that needs explanation, it will make more sense why they are included.).

3) For both the ATR-FTIR and ss-NMR experiments, the authors need to provide more details regarding sample preparation. Are solid precipitates isolated by centrifugation? Are KBr pellets made? Was the solid packed into the NMR tubes? Such sample preparation details are critical to understanding the results.

4) On page 17, line 388, please provide a reference for the "traditional polyelectrolyte-enhanced wetting of surfaces."

5) On page 7, line 154, please specify in which figure the ds-DNA macromolecules can be seen.

6) In the Methods section, please provide an item number for the track-etch membranes used from VWR. These membranes are notoriously supplier and item number specific, especially as it relates to the surface treatments, which can vary. More information is required to repeat the experiments as described.

7) Units:

a. Throughout the manuscript, lowercase l's are used for liter instead of uppercase L's. (e.g., g/l instead of g/L)

b. Units are not provided for molecular weights. Please specify g/mol

Response to Reviewers' comments:

We greatly appreciate the reviewers for the positive and inspiring comments on our manuscript. We have revised the manuscript according to the comments. All the revisions are yellow highlighted in the manuscript as well as in this response letter. The point-to-point responses to the reviewers' comments are as follows.

Reviewer 1:

Reviewer comments (General 1): The manuscript describes amorphous calcium carbonate precursors formed in presence of various polyelectrolytes, positive and negatively charged polyelectrolytes, (Pasp, PAA, DNA and PAH, with main focus on DNA). The authors challenge the concept of widely accepted precursor phase so called "PILP" (polymer induced liquid precursor). The authors argue that this specific phase is not a liquid at nano/microscale, despite of their macroscopic liquid-like behaviors observed by many researchers for many years. The authors convey their argument by using CryoTEM and NMR mainly. The results show the primary clusters formed in presence of the polyelectrolyte in the very early stage are ~2 nm in diameter, which don't coalesce as you would expect from liquid, therefore conclude the acronym PILP is misleading in some extent.

The quality of the manuscript is very good and comprehensive overall, but one of key characterizations, NMR data is only limited to DNA/CaCO₃ case, which is much less commonly accepted for PILP process, compared to ones with Pasp or PAA. It is a convenient choice as it allows to detect P-NMR signals, but in the weak side, this causes rather large discrepancy in concentrations (eg. 100 times more than reaction using other polyelectrolytes, 25mg/ml). More importantly, while the authors intend to compare their NMR data to the results from the previous NMR work from Gower group, I found that not only different polyelectrolytes, (DNA.PILP vs Pasp/PILP) were used, but also solution conditions and precipitation method are entirely different. I am not very convinced by the fact the two systems are equivalent to directly compare. So the main question still remains.

Author reply:

ds-DNA vs pAsp

We appreciate the reviewer for the very positive comments and kind suggestions. Indeed a much higher ds-DNA concentration (2.5 g/L) is required to induce PILP formation with 10 mM Ca²⁺. The structure and properties of ds-DNA induced PILP are however similar to the "classical PILP", e.g. induced by the original

Gower procedure using 25 mg/L pAsp and 10 mM Ca^{2+} . (see Text Ref. 20) This means similar film formation or nanorod growth in track etch membranes were observed in both system, and most of all, similar initial products (assemblies of ~ 2 nm sized ACC clusters) were formed as detected by cryoTEM. Since NMR spectroscopy only reflects the chemical change in short range (0.1 to 10 nm), we therefore feel comfortable to use the ds-DNA system for the NMR study of the PILP process.

Moreover, the ds-DNA system is the only system that can reasonably be used to monitor the PILP process by liquid state NMR. Typically the PILP process (e.g. the original Gower procedure) is induced through the $(\text{NH}_4)_2\text{CO}_3$ diffusion method, at a stable pH value (~ 10) with a low concentration of polymer (25 mg/L) and low supersaturation (10 mM Ca^{2+} and up to ~ 5 mM CO_3^{2-} , Ref: Ihli J, Bots P, Kulak A, Benning LG, Meldrum FC. Elucidating mechanisms of diffusion-based calcium carbonate synthesis leads to controlled mesocrystal formation. *Adv Funct Mater* **23**, 1965-1973 (2013).). In such a system, the only NMR measurement that would provide information about the reaction process could be ^{13}C liquid-state NMR. However, the ^{13}C content of natural or synthetic materials is only $\sim 1\%$. Moreover, the gyromagnetic ratio of ^{13}C (γ) is only 0.6 of that of ^{31}P . Since the measurement time for each reaction stage is also limited (< 30 min), it is technically not possible for us to perform a reliable ^{13}C liquid-NMR measurement in a typical PILP system, even with 100% ^{13}C enrichment.

It is noteworthy that the system used by the Gower group for NMR study (see Text Ref. 33) is also not a typical PILP system, probably due to the difficulty we mentioned above. In their study a titration method was used (rather than the $(\text{NH}_4)_2\text{CO}_3$ diffusion method), with a relatively high CO_3^{2-} concentration (30 mM), moderate pH (~ 8.5) and 100% ^{13}C enrichment. This method generates relatively stable solutions that do not precipitate after 18 hours, and thus allows the ^{13}C liquid-NMR measurements. The extra peak observed in the ^{13}C liquid-NMR was attributed to the presence of PILP phase, and was considered as a direct evidence of the liquidity of PILP, although it was not shown that typical products of PILP processes (thin film or nanorods) could form in such a system.

Comparison with NMR data from Gower group

Indeed the precipitation method we used is different from the NMR work of Gower group (see Text Ref. 33), on which we should have been more clear in the manuscript. The aim of our discussion about the work of Gower et al. however is not to directly compare these two results and demonstrate they are the same, but to show that the NMR results of Gower et al. do not necessarily demonstrate a liquid-liquid phase separation, but could also be explained by the formation of ~ 2 nm clusters as we observed.

Revisions made to the manuscript:

We have clarified the experimental differences between our work and the NMR work of Gower group, and emphasized the aim of the discussion in the revised manuscript:

On Page 16, Line 370-374: Apparently, our results contrast with the **NMR** work of Bewernitz et al. (Text Ref. 33) who used a ^{13}C enriched titration system with high $\text{CO}_3^{2-}/\text{HCO}_3^-$ concentration, moderate pH (~ 8.5) and presence of pAsp 18 mg/L). An extra ^{13}C liquid NMR signal was detected in this system after

Ca²⁺ titration which they attributed to PILP, although it was not shown that typical products of PILP processes (thin film or nanorods) could form in such a system.

On Page 17, Line 379-382: Hence, our results are consistent with previous reports in spite of the differences in experimental conditions, and all data can be interpreted such that PILP is a dispersion of dynamic assemblies of ~2 nm sized ACC clusters that are cross-linked by charged polymers, without necessarily the presence of a liquid-liquid phase separation.

Reviewer comments (General 2): In the same line, the authors emphasize generality of their observations, but overlook differences from specific cases. For example, while Pasp and DNA cases appear similar, PAA and PAH don't, by judging the size or structures of aggregates (figure 2 vs figure s8)

Author reply: We thank the reviewer for the comment, and realize that we have not been clear enough on this issue. First of all, the images shown in the original Fig. S8 d-i may cause confusion. These are not showing individual PILP nanoparticles, but a liquid-like complex of pAH and CO₃²⁻, indicating the coacervation of these objects, and are not to be compared with the images of PILP from pAA or pAH. To generate more clarity we have made two separate supplementary figures (Fig. S13 and S14) for these images.

Meanwhile, indeed the PILP nanoparticles induced by different polymers are different in sizes (from 30 to 200 nm) and shapes (e.g., the particles induced by pAH or ds-DNA seem to be more spherical than those formed with pAsp or pAA). In spite of these differences which may result from the different properties of the polymers (e.g., molar weight, charge density, length, etc.), however, all of the PILPs are colloidal nanoparticles assembled from ~2 nm sized subunits. So the PILPs are essentially have the same build-up at the nanoscale, and are colloidal nanoparticles instead of liquid droplets since they do not coalesce with each other, and the ~2 nm clusters inside them also do not coalesce. This is also what we want to highlight in this report. Still, we agree with the reviewer that the differences in the PILP nanoparticles induced by different polymers should not be overlooked. We have now addressed this in the manuscript.

Revisions made to the manuscript:

1. We have made two separate supplementary figures (Fig. S13 and S14) for the original Fig. S8.
2. We have modified the text accordingly, and described the difference of PILP induced by different polymers:

On Page 7-8, Line 170-178: Similar assemblies of ~2 nm subunits were found in the early stages of the pAA or pAH (25 mg/L) induced PILP processes (see Supplementary Fig. S13), underlining the universality of our observations. It is noteworthy that the assemblies induced by different polymers are different in sizes (~200 nm sized assemblies could be observed for pAH) and shapes (the particles induced by pAH or ds-DNA seem to be more spherical than those formed with pAsp or pAA). These differences which may result from the different properties of the polymers (e.g., molar weight, charge density, length, etc.). Meanwhile, when pAH was used at a higher concentration (1000 mg/L), indeed droplet-like objects were observed as was originally reported by Cantaert et al (see Supplementary Fig. S14).

Reviewer comments (General 3): I looked up the original statement about PILP, and found out in many papers, the phase has already been described as "liquid-like" behavior. Therefore, the closing remark of

this paper “we therefore suggest that the abbreviation PILP for polymer-induced liquid-like precursor” would not convey any strong message as the authors wish. I suggest to reconsider it.

Author reply: We agree with the reviewer on this suggestion.

Revisions made to the manuscript:

We have revised the closing mark accordingly:

On Page 19, Line 444-448: We therefore suggest to update the abbreviation of PILP into PINP for polymer-induced nanoparticulate precursor, highlighting its nanoparticulate microstructure. The presented model for PILP (PINP) not only aids to our understanding on how control over crystal morphology can be achieved in materials of technological relevance, it may also provide mechanistic insights into the formation processes of minerals in biological systems.

Reviewer comments (Specific 1): Assuming those aggregates (very much uniform sizes) are coming from 2 nm clusters, one would expect to see many individual 2 nm clusters (single/double/triple...) in surrounding media. I found this most strange. This makes me think whether the very early stage was missed. From schematic/mechanism of the process, should Ca/polyelectrolyte complexes should be the first products? Followed by formation of many individual 2 nm clusters ? The authors also mentioned on page16 “PILP consists of assemblies of 2 nm sized ACC clusters that are physically cross-linked by charged polymers, and no other product was detected before the formation of this phase in our experiment”

Author reply:

Individual clusters or small assemblies

We thank the reviewer for this inspiring question. Indeed, as shown by our cryoTEM/tomographic results in Figure 2, no individual ~2 nm cluster could be observed within the surrounding media when the 30-50 nm sized assemblies of these clusters are formed. Also no individual cluster, or smaller sized assembly, was observed in earlier products. To further confirm whether the invisibility of individual clusters is due to a lack of contrast, we applied the early stage products (15 min for ds-DNA experiment and 90 min for pAsp experiment) on graphene oxide (GOx) coated cryoTEM grids. This recently developed method allows the formation of much thinner vitrified ice layers (~10 nm, See Text Ref. 43), which greatly improves the contrast of cryoTEM images and benefits the detection of ~1 nm sized clusters. Still no individual clusters or smaller assemblies were detected in these experiments as shown by Fig. R1, suggesting that their concentration is negligible in our experiments.

We can understand the extremely low concentration of individual clusters or smaller assemblies taking into account the high surface area ratio of the clusters and the instability of the ACC phase in water solution. As a result, in our experiments most of the individual clusters will immediately dissolve unless they are nucleated on, or bound with the charged polymers, and assemble into the >30 nm sized

assemblies. Given this, an extremely low equilibrium concentration is expected for the individual clusters or smaller assemblies in our experiments.

Figure R1. CryoTEM images of early stage reaction solutions prepared on GOx supported TEM grids. (a-b) Reaction solution with 25 mg/L of pAsp after 90 min of experiment. (a) shows the overview of the ice layer. The SAED pattern in the inset shows the diffraction signal of GOx monolayer, or several layers in the same orientation. (b) shows the zoom-in image, which shows only noise-like signal similar to the flat-field (empty) image as shown in (c). (d-e) Reaction solution with 2.5 g/L of ds-DNA after 15 min of experiment. (d) shows the overview of the ice layer. The SAED pattern in the inset shows the diffraction signal of GOx layers in two slightly misaligned orientations. (e) shows the zoom-in image, which shows densely packed ds-DNA molecules. The higher density of ds-DNA is due to the 60 seconds of waiting time applied during the vitrification, which allows the ds-DNA molecules to condense on the GOx layers. (b) and (e) are both taken with a defocus value of $-1.5 \mu\text{m}$. Scale bars: (a, d) 500 nm, insets of (a, d) 2 nm^{-1} , (b, c, e) 20 nm.

Revisions made to the manuscript:

1. The cryoTEM images of earlier stages on GOx coated TEM grids (Fig. R1) are added into the supplementary information as Fig. S10.
2. The corresponding experimental method was added into the Methods part:

On Page 21, Line 501-506: To study the early stage samples, 3 μL of reaction solution samples were applied on a GOx coated TEM grid (Text Ref. 43), and 20% (v/v) IPA in ultrapure water was used in humidifier. After 60 s of waiting, the grid was blotted for 3 s and vitrified. CryoTEM imaging was performed under $\sim 3 \mu\text{m}$ defocus on a FEI-Titan TEM equipped with a field emission gun (FEG) and operating at 300 kV. For samples on GOx coated grids, imaging was done using a parallel beam with an illuminated area of 670 nm (nanoprobe), with a defocus value of $-1.5 \mu\text{m}$.

3. Fig. S10 is referred in the manuscript together with Fig. S9:

On Page 6-7, Line 140-142: The earliest products were observed after ~ 150 min of growth, which are nanoparticles (NPs) with diameters of ~ 50 nm (Fig. 2a, see also Supplementary Fig. S9-10 for the cryoTEM images of even earlier stages).

4. The explanation for the low concentration of individual clusters or assemblies is inserted into the Discussions part:

On Page 6-7, Line 389-392: Without being stabilized by the polymers and assembled into the >30 nm sized assemblies, these ACC clusters will immediately dissolve due to their high surface area ratio and the instability of ACC phase in water solution, leading to an extremely low equilibrium concentration of the individual clusters or smaller assemblies in our experiments (see Supplementary Fig. S9-10).

Ca/polyelectrolyte complexes

And indeed as shown in Fig.6, the Ca^{2+} /polyelectrolyte complex is expected to be the first product, which actually should already form upon mixing the CaCl_2 and polymer solutions. The cryoTEM images from these complexes are however not expected to be very different from those of the free polymers, unless they aggregate and phase separate from the solution, such as is the case for the $\text{pAH}/\text{CO}_3^{2-}$ complex. This was not observed for the other charged polymers used in our experiments (ds-DNA, pAsp and pAA). The presence of these complexes could still be reflected by the binding between Ca^{2+} and ds-DNA as shown in the supplementary Table S5 and S6.

Revisions made to the manuscript:

We have supported our description of Ca^{2+} /polymer complex formation using Table S5 and S6:

On Page 18, Line 411-412: Taking negatively charged polymers as an example (Fig. 6a), the reaction starts with the formation of a Ca^{2+} /polymer complex (stage 1, see also supplementary Table S5 and S6).

Reviewer comments (Specific 2): The concentrations of polyelectrolytes are too low to stabilise all the 2 nm clusters. Taking Pasp (Mw 10000) for an example, it is only 1-2 μM , which is extremely low to stabilize mineral phase (10 mM) used here ?

Moreover as the authors claim, If the wet phenomenon were due to polyelectrolyte stabilizing the clusters, then demonstration of same phenomenon with the same polyelectrolyte stabilized (solid)

colloids would be the strongest and direct evidence? PAH or PAA are widely used to stabilize colloidal metal/inorganic nanoparticles in similar sizes, like Au clusters.

Authors could benefit from discussion further regarding “nanofluid” behavior : Generalized Route to Metal Nanoparticles with Liquid Behavior. *J. Am. Chem. Soc.* 128, 12074–12075 (2006).

Author reply:

The authors thank the reviewer for the questions, which greatly helped us to further improve the paper. As for the several points raised here:

ACC clusters stabilized by pAsp

Indeed the concentration of pAsp we used is only $\sim 2.5 \mu\text{M}$, which however means $\sim 0.22 \text{ mM}$ of aspartic acids ($-\text{COO}^-$ groups). According to a detailed compositional study performed by Gower et al. on pAsp-induced PILP (Ref: Dai L, Douglas EP, Gower LB. Compositional analysis of a polymer-induced liquid-precursor (PILP) amorphous CaCO_3 phase. *J Non-Cryst Solids* **354**, 1845-1854 (2008).), the $\text{Ca}^{2+}/\text{COO}^-$ ratio is $\sim 10:1$ for PILP induced by 20 mg/L of pAsp. Thus the $2.5 \mu\text{M}$ of pAsp used in our experiments should be able stabilize PILP representing 2.2 mM of Ca^{2+} . On the other hand, the 10 mM of Ca^{2+} ions in the reaction solution are not expected to be all transformed into the PILP phase, at least not at the same time. The actual concentration of PILP in our experiments is difficult to be accurately measured. However, the PILP products we got after centrifugal separation and drying is only $2\sim 3 \text{ mg}$ for each experiment (25 mL of reaction solution), which correspond to only to $\sim 1.2 \text{ mM}$ of Ca^{2+} in the reaction solution. Therefore, the 25 mg/L (or $\sim 2.5 \mu\text{M}$) of pAsp used in our experiments should be sufficient to stabilize all the $\sim 2 \text{ nm}$ ACC clusters in the PILP phase.

The high $\text{Ca}^{2+}/\text{COO}^-$ ratio in pAsp-induced PILP could be explained by the fact that the polyelectrolytes are binding only with the surface Ca^{2+} of $\sim 2 \text{ nm}$ sized ACC clusters, instead of all the Ca^{2+} ions in PILP. According to a recent computational study, each $\sim 2 \text{ nm}$ sized ACC cluster consists of $\sim 20 \text{ Ca}^{2+}/\text{CO}_3^{2-}$ ion pairs (See Text Ref. 60). The $10:1 \text{ Ca}^{2+}/\text{COO}^-$ ratio therefore suggests that each ACC cluster is bound with two COO^- groups in pAsp-induced PILP. Interestingly, a similar $\text{Ca}^{2+}/-\text{PO}_4^-$ ratio ($10:1$) was detected for the ds-DNA induced PILP by the ICP-OES measurements (see the newly added supplementary Table S2), and it remains unclear why a much higher concentration of ds-DNA (2.5 g/L) is required to induce the PILP process. We have incorporated part of this discussion into the manuscript.

Revisions made to the manuscript:

1. The following literature is added into the reference list as Ref. 61:

Dai L, Douglas EP, Gower LB. Compositional analysis of a polymer-induced liquid-precursor (PILP) amorphous CaCO_3 phase. *J Non-Cryst Solids* **354**, 1845-1854 (2008).

2. One paragraph of discussion is added into the manuscript:

On Page 16, Line 357-369: The fact that PILP is assembled from $\sim 2 \text{ nm}$ sized ACC clusters also explains why they could be stabilized by a relatively low concentration of polyelectrolytes e.g., 25 mg/L of pAsp, which contains only $\sim 0.22 \text{ mM}$ of COO^- groups. Instead of binding with all the Ca^{2+} ions, the polyelectrolytes only need to bind with the surface Ca^{2+} of the $\sim 2 \text{ nm}$ sized ACC clusters in PILP. Each of these ACC clusters consists of $\sim 20 \text{ Ca}^{2+}/\text{CO}_3^{2-}$ ion pairs according to a recent computational study (Text Ref. 60). As a result, much more Ca^{2+} could be stabilized by one COO^- group. Indeed, as shown by a

previous compositional study of PILP by Dai et al., the $\text{Ca}^{2+}/\text{COO}^-$ ratio is $\sim 10:1$ for PILP induced by 20 mg/L of pAsp, (Text Ref. 61) suggesting that each ACC cluster is bound with two COO^- groups in pAsp-induced PILP. As a result, 25 mg/L of pAsp could stabilize PILP representing ~ 2.2 mM of Ca^{2+} , allowing a higher concentration of PILP formation. Interestingly, a similar $\text{Ca}^{2+}/\text{-PO}_4^-$ ratio (10:1) was detected for the ds-DNA induced PILP by the ICP-OES measurements (see supplementary Table S2), and it remains unclear why a much higher concentration of ds-DNA (2.5 g/L) is required to induce the PILP process.

“Wetting ability” of PILP

Author reply:

We apologize for being not accurate on the discussion about the wetting ability of PILP, which is actually the ability of PILP nanoparticles to attach to surfaces. For this to happen, the PILP nanoparticles must first be stabilized by the charged polymers, otherwise they will crystallize into large crystals, rather than attaching to surfaces and forming non-equilibrium morphologies such as thin films and rods. In that sense, the polyelectrolytes are essential for the “wetting phenomenon” since they clearly play a vital role in stabilizing the PILP particles

However, one requirement for the “wetting” is the affinity between PILP particles and substrates. This is not determined only by the properties of the polymers on the surface of PILP particles, as we mistakenly stated in the original manuscript. Indeed, PILP induced by different polymers leads to thin film products with differences in thickness and surface roughness on the same substrate, as shown by our experimental results using pAsp, ds-DNA, pAA or pAH, as well as by observations in previous reports (Ref: Gower LB. Biomimetic model systems for investigating the amorphous precursor pathway and its role in biomineralization. *Chem Rev* **108**, 4551-4627 (2008).). This suggests that the affinity of PILP nanoparticles to surfaces is at least partly regulated by polymers. However, the interaction between colloidal particles and the surfaces is a complicated collective effect of many forces including electrostatic interaction forces, van der Waals forces, entropic forces and steric forces (Ref: Belloni L. Colloidal interactions. *J Phys: Condens Matter* **12**, R549 (2000).). As a result, the interaction should also be effected many other factors such as the size and chemical composition of the colloidal particles, the way polymers interact with the colloidal particles, as well as the ionic strength, pH value, etc. This has also been suggested in the PILP system by a previous report, showing that the “wetting” of PILP on different substrates could be drastically tuned with presence of 30 mM of Mg^{2+} (see Text Ref.35). It was proposed that the Mg^{2+} might influence the hydration level of PILP, but the detailed mechanism remains unknown.

Furthermore, PILP is not exactly the same with typical colloidal particles stabilized by surface polymers. They are actually 30-50 nm sized bicontinuous structures assembled by ~ 2 nm ACC clusters and polymers. These assemblies could grow larger and later on transform into crystals. Thus the interaction of PILP with different surfaces could be even more complex. As a result, the experiments of Au or other metal clusters stabilized by the same polymers would unlikely provide any clear evidence about why the PILP particles has a strong affinity to certain surfaces. To fully understand how the affinity of PILP to different surfaces is regulated by different polymers still require systematic experiments in the future, examining the PILP induced by polymers with different molar weight, charge density, diameter, length, etc., and how they interact with different surfaces, which is beyond our current discussions.

Revisions made to the manuscript:

We have corrected our inaccurate statement that the surface property of PILP is only determined by the polymer:

On Page 17, Line 399-400: The PILP NPs will completely cover surfaces that match with their ACC/polymer surfaces (or not at all when the surfaces are incompatible with the surface of PILP).

Reference for the liquid-like behavior

Author reply:

We agree with the reviewer on this suggestion.

Revisions made to the manuscript:

1. We have added the following literature into the reference list as Ref. 64:

Warren SC, Banholzer MJ, Slaughter LS, Giannelis EP, DiSalvo FJ, Wiesner UB. Generalized Route to Metal Nanoparticles with Liquid Behavior. *J Am Chem Soc* **128**, 12074-12075 (2006).

2. Ref.64 is described accordingly in the discussion about the liquid-like behavior of PILP:

On Page 17, Line 395-396: Similar liquid-like behavior has been reported for surfactant functionalized colloidal particles (Text Ref. 63, 64). Due to steric repulsion of neighboring functional groups, these colloidal particles could flow even after drying (Text Ref. 64).

Reviewer comments (Specific 3): Line133 page 7 “did not coalesce to form continuous spherical objects” if I understood correctly, f and i in Figure S8 represent “liquid” nature of objects. However, even those doesn’t look very spherical, even though I do agree with the authors the absence of internal subunits. Out of curiosity, “nanoscopic fluidity” has ever been directly observed by in-situ EM?

Author reply:

Liquid droplets visualized by cryoTEM

We appreciate the reviewer for pointing this out. The f and i in original Figure S8 indeed show liquid-like objects that are pAH/CO₃²⁻ complex, as we mentioned in the reply to the General Comment 2. However, we realized that our previous description that liquid droplets will always be spherical under cryoTEM was not rigorous. In the ideal case without considering gravity and small local inhomogeneities, the equilibrium shape of liquid droplets dispersed within another liquid indeed should be spherical (Ref: Dubochet J, *et al.* Cryo-electron microscopy of vitrified specimens. *Q Rev Biophys* **21**, 129-228 (1988)). This is in order to minimize their surface area and reduce the liquid-liquid interfacial energy. For the droplets with smaller size (higher surface area ratio) and/or high liquid-liquid interfacial energy, their shape will be strongly regulated by the interfacial energy and thus close to spherical (Figure R2a). This has indeed been observed for oil nanodroplets dispersed in water, which are continuous and spherical objects under cryoTEM (Ref: Klang V, Matsko NB, Valenta C, Hofer F. Electron microscopy of nanoemulsions: an essential tool for characterisation and stability assessment. *Micron* **43**, 85-103 (2012).). For droplets with bigger size (lower surface area ratio) and/or lower liquid-liquid interfacial energy, however, the shapes will be more easily disrupted by local fluctuation in density, temperature or ionic strength, etc. As a result, although still being continuous objects with smooth edges, their shape will become irregular (Figure R2b). This was observed for the pAH/CO₃²⁻ complex droplet in original

Figure S8 f and I, which are micron-sized continuous objects with irregular shapes. The results suggest that the interfacial energy between the pAH/CO₃²⁻ complex and the solution is relatively low. Also these droplets are relatively larger (~microns). As a result, the shape of these droplets are less controlled by the interfacial energy, and their shapes are more irregular.

For the PILP phase, the 30~50 nm particles are basically spherical, but they consist of ~2 nm subunits, and occasionally are hollow (Figure 2). Furthermore, the micron sized aggregates of these particles also clearly show the nanoparticulate features as well as the ~2 nm subunits (Figure 2c and f), which are quite different from the pAH/CO₃²⁻ droplets (supplementary Figure S15). As a result, we conclude that the PILP is not a liquid.

It is noteworthy that the above mentioned features observed in cryoTEM (continuous, spherical or cloud like) can be associated with the liquid-like nature of an object, but do not prove it. For example, similar features have also been observed for soft gels in cryoTEM (Ref: Novoa-Carballal R, Pergushov DV, Müller AH. Interpolyelectrolyte complexes based on hyaluronic acid-block-poly (ethylene glycol) and poly-L-lysine. *Soft Matter* **9**, 4297-4303 (2013).). This is also why we only describe the objects in the original Fig. S8 i and f as “liquid-like”.

In situ EM

We assume the referee refers to *in-situ* liquid phase EM. This technique indeed allows the direct observation of nanoscopic fluidity, such as shown by the pioneering work of Lu et al. (Ref: Lu J, Aabdin Z, Loh ND, Bhattacharya D, Mirsaidov U. Nanoparticle dynamics in a nanodroplet. *Nano Lett* **14**, 2111-2115 (2014).). In this work, 3-10 nm sized gold nanoparticles encapsulated by ~30 nm sized water droplets on a flat solid surface were observed by in-situ liquid phase TEM. The observed nanodroplets are continuous objects with a close to spherical shape, matching with the cryoTEM observations. During the observation, the droplets flows and its shape deforms, accompanied by the movement of the gold nanoparticles within. Unfortunately, very high electron dose rate is usually required for high resolution *in-situ* liquid phase TEM observations (for the above mentioned work the electron dose rate is 2000 to 5000 e/(Å²-s)). Thus for electron beam sensitive materials such as ACC or polymers, the resolution of in-situ liquid phase TEM technique is still limited, and therefore it was not used for current study.(see Ref: **1.** Nielsen MH, Aloni S, De Yoreo JJ. In situ TEM imaging of CaCO₃ nucleation reveals coexistence of direct and indirect pathways. *Science* **345**, 1158-1162 (2014). **2.** Patterson JP, Proetto MT, Gianneschi NC. Soft nanomaterials analysed by in situ liquid TEM: Towards high resolution characterisation of nanoparticles in motion. *Perspectives in Science* **6**, 106-112 (2015).),

Figure R2. Scheme for the morphological difference of liquid droplets dispersed within another liquid. (a) For droplets with small size and/or high liquid-liquid interfacial energy, their shape will be close to spherical in order to minimize the surface area and

reduce the interfacial energy. (b) For droplets with large size and/or low liquid-liquid interfacial energy, their shapes will be more easily disrupted by the local inhomogeneity, and thus are more irregular.

Revisions made to the manuscript:

1. We have added Figure R2 together with the discussions on the liquid droplet morphologies into the supplementary information as Figure S1.
2. We have revised our text, clarifying that the liquid droplets in cryoTEM are continuous objects with smooth edge, as also asked by the Comment 1 of Reviewer 3:

On Page 2, Line 44-46: The PILP droplets are expected to be continuous objects with smooth edges under cryoTEM (see also supplementary Fig. S1). (Text Ref. 21, 30, 31)

On Page 7, Line 144-146: The NPs subsequently grew in size and aggregated to form larger structures, but did not coalesce to form continuous objects with smooth edges as would be expected for liquid droplets (Fig. 2b).

3. The following literature is added into the reference list as Ref. 31 and supplementary reference list as SI Ref. 1:

Dubochet J, *et al.* Cryo-electron microscopy of vitrified specimens. *Q Rev Biophys* **21**, 129-228 (1988).

Reviewer comments (Specific 4): In Figure 1. The crystalline platelet is formed by dissolution of ACC films, it contradicts again with the suggested mechanism/schematic here and the original hypothesis of PILP, where solid state transformation is implied?

Author reply:

“solid-transformation” vs “ACC dissolution”

In the PILP systems the first crystalline CaCO₃ plate usually forms within the ACC thin films. This is also observed in our experiments and may be due to a solid state transformation process, or to a dissolution-reprecipitation process, currently there is no evidence for one or the other in PILP systems. This is why the term “solid-transformation” was not used in our manuscript.

The next step is the growth of the plate through the dissolution of the amorphous film. This is a well know phenomenon in ACC-crystal transformation and is driven by the different solubilities of the two phases where calcite acts as a sink for the precipitation of Ca²⁺ and CO₃²⁻ ions.

Although this does not contradict with the formation of the first plate through a solid state transformation hypothesis, the latter cannot describe the entire transformation as the ACC phase is hydrated (usually ~1:1 CaCO₃ : H₂O, Ref: Ihli J, *et al.* Dehydration and crystallization of amorphous calcium carbonate in solution and in air. *Nat Commun* **5**, 3169 (2014).), and the polymers within PILP will be excluded after crystallization. Therefore a larger volume of PILP has to be consumed to form the same molar amount of dehydrated crystalline polymorphs such as calcite or vaterite, regardless of the

transformation pathways. This then must lead to a dissolution of the ACC thin film around the growing crystalline platelets.

Moreover, the overall supersaturation of the solution is decreasing during the experiment, as shown by our $[\text{Ca}^{2+}]$ profile. This will further promote the dissolution of ACC due to its relatively higher solubility comparing with calcite or vaterite.

Hence, the ACC thin film *should* dissolve at the same time when the calcite/vaterite platelets form, especially around the front of crystallization where ACC is being faster consumed. This is consistent with the previous report of Aizenberg et al. (see Text Ref. 59), and does not contradict with the possible solid state transformation process.

Revisions made to the manuscript:

1. We have added the following literature into the reference list as Ref. 65:

Ihli J, et al. Dehydration and crystallization of amorphous calcium carbonate in solution and in air. *Nat Commun* 5, 3169 (2014).

2. We have clarified why the ACC thin film should dissolve in the Discussions part:

On Page 18, Line 418-423: Nucleation of crystallized CaCO_3 happens later, generating crystalline plates within the thin film (stage 5, see also Fig. 1f). Since the ACC phase is hydrated (usually $\sim 1:1 \text{ CaCO}_3 : \text{H}_2\text{O}$) (Text Ref. 65) and the polymers within PILP will be excluded after crystallization, a larger volume of PILP is consumed to form the same molar amount of dehydrated crystalline polymorphs such as calcite or vaterite. This then lead to a dissolution of the ACC thin film around the growing crystalline platelets, which is promoted by the decreasing supersaturation of the solution (Fig. 4d) due to the relatively higher solubility of ACC comparing with calcite or vaterite. After complete crystallization (stage 6, see also Fig. 1b), the majority of polymers are excluded from the crystals, while some of them become occluded.

Reviewer comments (Specific 5): NMR: the pH is not constant during the reaction, changes in ionic strength and pH in the solution will heavily affect the interaction and conformation of DNA and might diminish or enhance such effects. Therefore it is not clear that how to decouple the effect of interaction to ACC from pH effects, unless data of DNA/Ca at varied pH support otherwise.

Author reply:

Liquid NMR measurements: the effect of pH and $[\text{NH}_4^+]$

We agree with the suggestion of the reviewer, and have investigated the effect of pH value change to our ^{31}P liquid NMR measurements. Only a slight fluctuation of the chemical shift (~ 0.02 ppm) and peak width (~ 0.04 ppm) was observed when the pH value of our reaction solution (with 10 mM CaCl_2 and 2.5 g/L ds-DNA) was raised from 5.3 to 10.0 by 50 mM NaOH, as shown in Figure R3a. This very small fluctuation may in fact be due to the intrinsic experimental variability of the measurements. So the pH will not have a significant effect on the chemical shift of ^{31}P signal during the reaction.

Nevertheless, we should consider that the generation of NH_4^+ ions due to the in-diffusion of NH_3 may influence the conformation of ds-DNA by changing the ionic strength. We investigated this by adding 5 to

100 mM of NH_4Cl into our reaction solution (Figure R3b). The addition of NH_4Cl indeed first shifts the ^{31}P signal to high field reaching a shift of $\Delta\delta = -0.06$ ppm at $[\text{NH}_4^+] = 10$ mM. When the concentration was further increased to 100 mM - which is close to the actual NH_4^+ concentration in the reaction solution (Ref: Ihli J, Bots P, Kulak A, Benning LG, Meldrum FC. Elucidating mechanisms of diffusion-based calcium carbonate synthesis leads to controlled mesocrystal formation. *Adv Funct Mater* **23**, 1965-1973 (2013).)- the signal shifted back to 0.05 ppm down field of the initial solution.

We therefore attribute the small down field chemical shift observed after 200 min of the reaction (Fig. 4a and b) of 0.07 ppm (compared to the neat ds-DNA solution) mainly to the increase of the ionic strength and the influence of NH_4^+ on the conformation of ds-DNA.

Beyond this small shift, our results show a significant high field shift ($\Delta\delta = -0.14$ ppm) and broadening (0.27 ppm) of the ^{31}P signal of ds-DNA upon mixing with CaCl_2 , and then a continuous down field shift (0.21 ppm) and sharpening (0.27 ppm) of the signal during the diffusion experiment. This trend is unlikely due only to the in-diffusion of NH_3 , and should still be mainly attributed to the release of Ca^{2+} from binding with ds-DNA due to CaCO_3 formation.

Figure R3. ^{31}P liquid state NMR measurement of the solution containing 10 mM CaCl_2 and 2.5 g/L ds-DNA measured, with different pH values and $[\text{NH}_4^+]$. (a) Spectra of the solutions with pH=5.3, 7.0, 8.0, 9.0 and 10.0, respectively. The pH value was adjusted by 50 mM NaOH. (b) Changes of chemical shift and peak width of the ^{31}P signals upon the increase of pH value (compared with the original solution, pH=5.3). (c) Spectra of the solutions with $[\text{NH}_4^+]=0, 5, 10, 20, 50$ and 100 mM, respectively. The $[\text{NH}_4^+]$ was introduced by mixing with NH_4Cl solutions. (d) Changes of chemical shift and peak width of the ^{31}P signals upon the increase of $[\text{NH}_4^+]$ (compared with the original solution, $[\text{NH}_4^+]=0$).

Revisions made to the manuscript:

We have added Figure R3 and the corresponding discussions into the supplementary information as Figure S15, and modified the text at these two locations:

On Page 12, Line 269-271: The increase of pH value was found to have no significant effect to the free $[\text{Ca}^{2+}]$ and liquid ^{31}P NMR measurements, while slight decrease of zeta potential was detected due to deprotonation (see supplementary Fig. S15a-b, Table S5 and S6).

On Page 12, Line 321-323: After 200 min the peak showed a width similar to the neat ds-DNA solution, but with a lower chemical shift due to the increased ionic strength related to the introduction of NH_4^+ and CO_3^{2-} ions in the solution (see supplementary Fig. S15c-d).

Reviewer comments (Minor 1):

Line 62 page 3 “with unprecedented detail the formation and transformation”
I would not agree with this.

Author reply: We agree with the reviewer on this suggestion.

Revisions made to the manuscript:

We have revised this sentence:

On Page 3, Line 68: Here we present an in-depth investigation of the formation and transformation processes of PILP

Reviewer comments (Minor 2):

Line 100 Page 4 “~extending over centimetres” all figures shown only show micron-meter scale, I believe it would be typo.

Author reply: We are sorry for this mistake. The thin films are usually several millimeters large, but they are impossible to be several centimeters sized since the glass slide used as growth substrate is ~2 cm

sized. The low magnification SEM image below (Figure R4) shows a ~1 mm sized region of thin film.

Figure R4. Low magnification SEM image of a ACC thin film formed on the glass slide after 5 h of reaction with presence of 25 mg/L of pAsp. Scale bar: 200 μ m.

Revisions made to the manuscript:

We have corrected the mistake by changing “centimeters” into “millimeters” **on Page 4, Line 106.**

Reviewer comments (Minor 3):

Using unified units, hour or mins, will help readers

Author reply: We thank the reviewer for the kind suggestion. However, we used different time units on purpose. For the *in-situ* measurements (CryoTEM, liquid state NMR, pH/[Ca²⁺] curve, DLS/zeta potential measurement, etc.), we used “mins” since these measurements reflect the real time status of the reaction solutions. For the characterization on dry samples, the samples have to be taken out of the reactions solution, thoroughly washed by ethanol, and then dried in room temperature for overnight before measurements. Several minutes will be required before the reaction process is fully quenched by de-hydration, thus the time scale could no longer be as accurate as the *in-situ* measurements, and “hours” were used for these results. For this reason we prefer to keep the units unchanged.

Reviewer comments (Minor 4):

Supplementary figure s7 : in raman spectra (f), what do the humps around 500-600 and 800 cm⁻¹ come from, doesn't match with ACC ?

Author reply: We appreciate the reviewer for pointing this out. ACC was reported to show only one broad peak around 1085 cm⁻¹ in Raman measurements (Ref: Addadi L, Raz S, Weiner S. Taking advantage of disorder: amorphous calcium carbonate and its roles in biomineralization. *Adv Mater* **15**, 959-970 (2003).). Our Raman spectra measured on the ACC thin films, however, show 3 humps around 500-600, 800 and 1085 cm⁻¹. Indeed 3 similar humps are also detected from the glass substrates we used (Figure R5). For the ACC thin film samples, however, the hump around 1085 cm⁻¹ is significantly enhanced compared with the glass substrate, still indicating the presence of ACC in the thin films.

Figure R5. Raman spectra taken on the products grown on a glass substrate with 25 mg/L of pAsp or 2.5 g/L of ds-DNA. The spectrum of the glass substrate is also shown for comparison. The glass substrate shows 3 humps around 500-600, 800 and 1085 cm⁻¹, respectively. For the thin films grown with pAsp for 5 h or with ds-DNA for 2 or 5 h, the hump at 1085 cm⁻¹ increases significantly, suggesting the formation of ACC. (SI Ref. 3) The rhombic platelet (platelet 1) grown with pAsp and the platelet grown with ds-DNA found at 24 h showed peaks corresponding to the CO₃²⁻ v₁ symmetric stretch mode (1087 cm⁻¹), v₄ symmetric vibration mode (710 cm⁻¹) and external modes (154 and 282 cm⁻¹) of calcite. The round shaped platelet grown with

pAsp found at 24 h (platelet 2) showed peaks corresponding to the split CO_3^{2-} ν_1 symmetric stretch mode (1071 and 1087 cm^{-1}), ν_4 symmetric vibration mode (748 cm^{-1}) and external mode (300 cm^{-1}) of vaterite. (SI Ref. 4,5)

Revisions made to the manuscript:

1. We have added the signal of glass substrates to supplementary Fig. S3 and S12f.
2. The following literature is added into the supplementary reference list as SI Ref. 3:

Addadi L, Raz S, Weiner S. Taking advantage of disorder: amorphous calcium carbonate and its roles in biomineralization. *Adv Mater* **15**, 959-970 (2003).

Reviewer comments (Minor 5):

Supplementary table 5 : measurement at actual reaction pH (-9-10) would be more informative.

Author reply: We appreciate the reviewer for the suggestion. The original supplementary Table S5 is re-measured at pH=10.0 (Table R1), which shows only small deviations (~ 0.1 mM) to the previous results. This suggests the binding between Ca^{2+} and ds-DNA is relatively strong and thus not affected by the protonation/deprotonation of the phosphate groups. The zeta potentials in original supplementary Table S4 are also re-measured at pH=10.0 (Table R2), which are slightly lower than the values measured at pH=5.3 due to the deprotonation at higher pH.

Table R1. Free $[\text{Ca}^{2+}]$ measurement of 10 mM CaCl_2 solution containing different concentrations of ds-DNA. The phosphate group concentration was also estimated from the molecular weight (~ 225000) and number the number of base pairs (~ 300) of the ds-DNA. The measurements were performed at pH=5.3 and 10.0, respectively. It shows that the free Ca^{2+} in the solution decreased with increased ds-DNA concentration. With 2.5 g/L ds-DNA, only 7.25 mM of free Ca^{2+} was detected in the solution with pH=5.3, indicating 2.75 mM of Ca^{2+} was bound to ds-DNA. The free $[\text{Ca}^{2+}]$ detected at pH=10.0 only deviate slightly with the values detected at pH=5.3, suggesting the binding between Ca^{2+} and ds-DNA is relatively strong and thus not significantly affected by the protonation/deprotonation of the phosphate groups.

ds-DNA Concentration (mg/L)	Phosphate Group Concentration (mM)	Free $[\text{Ca}^{2+}]$ (mM), pH=5.3	Bound $[\text{Ca}^{2+}]$ (mM), pH=5.3	Free $[\text{Ca}^{2+}]$ (mM), pH=10.0	Bound $[\text{Ca}^{2+}]$ (mM), pH=10.0
0	0	10	0	10	0
25	0.07	9.65	0.35	9.67	0.33
66	0.18	9.57	0.43	9.52	0.48
250	0.67	9.26	0.74	9.31	0.69
666	1.77	8.85	1.15	8.97	1.03
2500	6.67	7.25	2.75	7.44	2.56

Table R2. Zeta potential measurement of 2.5 g/L ds-DNA solutions mixed with different concentrations of Ca^{2+} . The measurements were performed at pH=5.3 and 10.0, respectively. The zeta potential of pure ds-DNA solution was highly negative due to the negatively charged phosphate groups. The zeta potential significantly increased with increased Ca^{2+} concentration, indicating the binding between Ca^{2+} and ds-DNA. The zeta potentials measured at pH=10.0 are slightly lower than those measured at pH=5.3 due to the deprotonation at higher pH.

Ca^{2+} Concentration (mM)	Zeta Potential (mV), pH=5.3	Zeta Potential (mV), pH=10.0
0	-50.2 \pm 4.34	-53.1 \pm 6.01
1	-30.1 \pm 3.73	-33.9 \pm 3.99
2	-22.5 \pm 3.46	-26.2 \pm 3.68
5	-14.3 \pm 3.48	-20.3 \pm 4.26

10	-12.9±3.56	-15.9±5.56
----	------------	------------

Revisions made to the manuscript:

We have added the Table R1 and R2 into the supplementary information as Table S6 and S5, and described the effect of pH value to free $[Ca^{2+}]$ and zeta potential measurements in the text:

On Page 12, Line 269-272: The increase of pH value was found to have no significant effect to the free $[Ca^{2+}]$ and liquid ^{31}P NMR measurements, while slight decrease of zeta potential was detected due to deprotonation (see supplementary Fig. S15a-b, Table S5 and S6). This suggests that the interaction between ds-DNA and Ca^{2+} is relatively strong and not affected by the deprotonation of ds-DNA.

Reviewer comments (Conclusion):

This is thorough studies with many beautiful data, particularly the cryo –TEM images and videos are striking, I am sure that crystallization/biomineralization community could benefit from. Therefore I would encourage publication if the authors address the comments and questions, hoping it would strengthen the paper.

Author reply: We thank the reviewer again for the encouragement and fruitful discussions. We have modified the manuscript accordingly and believe its clarity has been significantly improved. We hope the manuscript now fits the standard for publication on Nature Communications.

Reviewer 2:

Reviewer comments (General):

The authors studied the PILP process using advanced analytical techniques such as CryoTEM, Raman, and NMR techniques. The study highlight the mechanism behind mineral formation process, which may be relevant to biomineral formation or synthesis of novel biomimetic materials. The work is important. However, some of the results and discussion may be more convincing after more analysis and review of prior literatures.

Author reply: We thank the reviewer for the positive comments, and have addressed the specific comments point-by-point as follows.

Reviewer comments 1:

Line 85-95: Figure 1 e-h and supplementary video 1 describes the formation of thin calcite crystals at the later stage. However, the reviewer highly doubt the dissolution of ACC film in regions over 200 um diameters resulted in the formation of the calcite crystal. It is highly possible that ACC film still there and polymer is completely consumed at the late stage, the remaining Ca (which may be released from dissolution ACC phase as well) will react with CO_3 to form Calcite crystals. The observed "dissolution" may be just an artifact of focusing of objective lens on the crystals. If calcite crystal formed at the expense of dissolution of ACC phase, one should observe the disappearance of ACC film close to the crystal first, then extend the disappearance to outside. In summary, the reviewer doubts the observed disappearance of ACC from outside to inside.

Author reply: We agree with the reviewer's viewpoint and this is indeed a misunderstanding. We intended to describe the process as outlined by the reviewer. Indeed the dissolution of the ACC thin film

should start around the crystals first, and extend outside to the film edge, as shown also in the supplementary Movie S1. The focusing of objective lens is not changing here, which could be reflected by the stable contrast of the crystalline edges.

Revisions made to the manuscript:

We have described the process more clearly in the figure caption of Figure 1:

On Page 4, Line 97-98: A crystalline platelet nucleated within the film in (f), and grew in (g) and (h). The film near the growth front of the crystalline platelet started to dissolve in (g). The dissolution extended outside to the film edge and the film was fully dissolved in (h).

Reviewer comments 2:

Line 246-251: The authors talked about the release of Ca from Ca-DNA complex, which is correct. However, when Ca is released from Ca-DNA, there is already CaCO₃ phase formation. The observed Z-potential change measurement is based on particles. At this stage, there are two types of particles that contribute to the Z-potential: ACC particles and maybe macromolecular DNA particles. Thus the change of Zeta-potential does not confirm the release of Ds-DNA.

Author reply: We thank the reviewer for pointing this out. Indeed the decrease of Z-potential could also be an effect of Ca²⁺/CO₃ binding. The release of ds-DNA could still be confirmed by our EDS, FTIR, NMR and ICP-OES measurements.

Revisions made to the manuscript:

We have corrected the text accordingly:

On Page 12, Line 268: During this process also the zeta potential decreased to -24.2 mV due to the partial release of ds-DNA and/or binding between Ca²⁺ and CO₃²⁻ (Fig. 4c).

On Page 14, Line 318: During crystallization a continued decrease of the zeta potential to -34.3 mV was observed, in agreement with the release of the ds-DNA/formation of CaCO₃ during the process (Fig. 4c).

Reviewer comments 3:

Line 110-111: Crystal growth at air-liquid interface and on glass slides substrates are different matters. Has Gower mentioned that 2 μm sized particles in the air/liquid interface formed the film on the glass slide? The film formation refers only to the CaCO₃ on glass substrates.

Author reply: We again humbly feel here is a misunderstanding. As written in our manuscript, during the in-situ OM observation of the thin film formation process, the 2 μm sized particles were observed at the glass/solution interface, and there was no air/liquid interface involved in these experiments (please also check supplementary Fig. S4 for details). In Gower's report on PILP, (Text Ref. 20) similar 2 μm sized particles/droplets were observed on the surface of air bubbles in the solution, which might be the air/liquid interface mentioned by the reviewer. However, later on these particles were also observed near the CaCO₃ thin film formed on the glass substrate. Based on this observation, it has been proposed by Gower et al. that the CaCO₃ thin film was formed by the coalescence of PILP "droplets".

Revisions made to the manuscript:

We have referred to Gower's work again after this sentence, making it clear that it was proposed by Gower et al. that the thin film was formed by coalescence of these 2 μm sized particles/droplets:

On Page 5, Line 121-122: However it was never observed as proposed by Gower et al. (Text Ref. 20) that these particles deposited and subsequently coalesced with the film.

Reviewer comments 4:

Line 317, Figure 6. The reviewer disagree with picture depicted in stage 6. Can the author proves there are free "excluded polymer" in the solution. During mineral formation, the polymer-Ca complex even release Ca to form CaCO₃, the polymer stills bounds to mineral surface and polymers imbedded in the mineral phase (the polymer-poor region forms crystal, the polymer-rich region remains ACC phase). The close association of polyaspartic acid with CaCO₃ mineral during and after the PILP process has been demonstrated by Gower's group by using the FITC-labeled Polyaspartic acid. Please confirm the existence of free polymer as the author demonstrated in Stage 6. There is a mechanism figure in the below reference which may be cited and discussed for comparison purpose. "Dai, L. J.; Cheng, X. G.; Gower, L. B., Transition Bars during Transformation of an Amorphous Calcium Carbonate Precursor. *Chemistry of Materials* 2008, 20, (22), 6917-6928."

Author reply: We thank the reviewer for the question. As shown by our FTIR, EDS, NMR and ICP-OES results, the ratios of ds-DNA in the crystallized samples are much lower than the PILP phase, suggesting that the majority of polymers has been excluded during the process. This matches not only with the observation of Aizenberg et al.(Text Ref. 59), but also was suggested by the paper of Gower et al. that was mentioned by the reviewer (Ref: Dai L, Cheng X, Gower LB. Transition bars during transformation of an amorphous calcium carbonate precursor. *Chem Mater* 20, 6917-6928 (2008).). Furthermore, in another compositional study on PILP of Gower et al., it has been confirmed that the polymers as well as water in PILP were excluded during the crystallization process.(see Text Ref. 61) Certainly some of the polymers could be trapped within the crystals, in some cases forming "transition bars", as mentioned by the reviewer. However, the transition bars were not observed in our experiments, as well as most studies of PILP as far as we know, suggesting this is not a common phenomenon in the PILP systems. However, one thing that was missing in the Stage 6 of Figure 6 is that the polymers should preferably attach to the surfaces of crystals, from where they were excluded.

Revisions made to the manuscript:

We have modified the State 6 of Figure 6, showing that the polymer preferably attach to the surface of crystals, and modified the text accordingly:

On Page 18, Line 423-425: After complete crystallization (stage 6, see also Fig. 1b), the majority of polymers are excluded from the bulk crystals to the crystal surfaces or back to the solution, while some of them become occluded.

Reviewer comments 5:

Line 443, EDS used to quantify the Ca:P:Cl ratio was not an accurate method and the quantification is only accurate on smooth surface and. A better method such as dissolving the material and measure by other technique including ICP-AES should be used

Author reply:

ICP-OES measurements

We agree with the reviewer on this suggestion. Inductively coupled plasma optical emission spectrometry (ICP-OES) is performed to more accurately measure the elemental ratio of the samples. The samples were dissolved in 5 mass % HCl and heated at 120 °C for 16 hrs before the measurement according to a previous report (Ref: Holden MJ, Rabb SA, Tewari YB, Winchester MR. Traceable

phosphorus measurements by ICP-OES and HPLC for the quantitation of DNA. *Anal Chem* **79**, 1536-1541 (2007).), in order to fully digest the ds-DNA molecules in the samples for a more accurate phosphorous measurement. Mass fractions of four elements (Ca, P, Mg and Na) are measured for each sample (Table R3). The results show similar trend of chemical composition change in different samples as indicated by the EDS measurements (Table R4), thus do not affect the discussions in the original manuscript.

Unfortunately, the sensitive spectral lines of chloride have wavelengths of around 135 nm, which fall in the vacuum ultraviolet region (<190 nm) and are difficult to be detected by conventional ICP-OES due to the low transmission of the vacuum UV light in atmospheric oxygen or water. As a result, we are unable to accurately measure the chloride fraction in the ds-DNA/Ca²⁺ complex using our ICP-OES spectrometer. Considering the similarity of the EDS and ICP-OES measurements and the thorough cleaning procedure that we used for the ds-DNA/Ca²⁺ complex (5 rounds of centrifuge inside 70% ethanol), we think the actual chloride ratio inside the complex should not be very high and thus will not significantly affect the solid-state NMR measurement results.

Table R3: The mass fractions of Ca, P, Mg and Na in the ds-DNA, the ds-DNA/Ca²⁺ complex and CaCO₃ products grown with presence of ds-DNA and slow stirring (100 rpm) measured by ICP-OES.

	Ca (mg of sample g ⁻¹)	Ca (mg of sample g ⁻¹)	P (mg of sample g ⁻¹)	P (mg of sample g ⁻¹)	Mg (mg of sample g ⁻¹)	Mg (mg of sample g ⁻¹)	Na (mg of sample g ⁻¹)
Absorbance Wavelength	396.85 nm	317.933 nm	177.495 nm	178.287 nm	279.553 nm	280.270 nm	279.553 nm
ds-DNA	---	---	71.52±1.05	71.51±0.95	15.25±0.45	15.20±0.41	27.34±6.73
	Average value: ---		Average value: 71.52		Average value: 15.23		
ds-DNA/Ca²⁺	40.45±0.21	40.15±0.25	61.62±2.92	61.13±2.55	0.93±0.04	0.92±0.04	---
	Average value: 40.30		Average value: 61.38		Average value: 0.93		
2 h CaCO₃	259.9±2.5	262.8±2.7	18.42±2.13	18.72±1.95	5.16±0.03	5.16±0.03	---
	Average value: 261.4		Average value: 18.57		Average value: 5.16		
3 h CaCO₃	349.1±7.8	351.7±7.3	4.57±0.28	4.46±0.28	4.84±0.01	4.85±0.01	---
	Average value: 350.4		Average value: 4.52		Average value: 4.85		
5 h CaCO₃	372.4±6.9	376.7±7.8	3.78±0.27	3.76±0.25	5.53±0.12	5.53±0.11	---
	Average value: 374.6		Average value: 3.77		Average value: 5.53		

Table R4: The atom ratios of different elements in the ds-DNA, the ds-DNA/Ca²⁺ complex and CaCO₃ products grown with presence of ds-DNA and slow stirring (100 rpm) measured by EDS and ICP-OES^a, respectively. In the Ca containing samples the ratio of Ca was set as 1, while in ds-DNA the ratio of P was set as 1 for the sake of comparison.

	Ca	P	Mg ^b	N ^c	Na	Cl
ds-DNA (EDS)	---	1	0.24	11.29	0.65	---
ds-DNA (ICP-OES)	---	1	0.272	N/A	0.521	N/A
ds-DNA/Ca ²⁺ (EDS)	1	2.28	---	22.31	---	0.14
ds-DNA/Ca ²⁺ (ICP-OES)	1	1.966	0.038	N/A	---	N/A
2 h CaCO ₃ (EDS)	1	0.16	---	3.00	---	---
2 h CaCO ₃ (ICP-OES)	1	0.092	0.032	N/A	---	N/A
3 h CaCO ₃ (EDS)	1	0.03	---	---	---	---
3 h CaCO ₃ (ICP-OES)	1	0.017	0.023	N/A	---	N/A
5 h CaCO ₃ (EDS)	1	0.03	0.03	---	---	---
5 h CaCO ₃ (ICP-OES)	1	0.013	0.024	N/A		N/A

a: For the elements that are measured at two wavelengths, the values are calculated according to the averaged value.

b: The ds-DNA powder contained small amount Mg²⁺ and Na⁺. Scarce Mg²⁺ was also detected in the CaCO₃ samples (5 h), due to the replacement of Mg²⁺ to Ca²⁺ in CaCO₃. (SI Ref. 17)

c: The expected N/P ratio in ds-DNA was ~3.75:1. The strong signal of N should be due to the relatively low accuracy of EDS when measuring light elements. (SI Ref. 18)

Revisions made to the manuscript:

1. The following literature is added into the reference list as Ref. 72:

Holden MJ, Rabb SA, Tewari YB, Winchester MR. Traceable phosphorus measurements by ICP-OES and HPLC for the quantitation of DNA. *Anal Chem* **79**, 1536-1541 (2007).

2. Table R3 is added into the supplementary information as Table S2. The original Table S2 is replaced by Table R4 as Table S3.

3. The experimental method of the ICP-OES measurements are added into the Methods part:

On Page 23-24, Line 585-600:

ICP-OES Measurements

Inductively coupled plasma optical emission spectrometry (ICP-OES) with end on plasma (Axial plasma) were performed using a SPECTROBLUE EOP spectrometer (AMETEK, Germany), with a measurement wavelength range of 165-770 nm. The generator of the spectrometer runs at 27.12 MHz, 1.4 kW. The samples were dissolved in 5 mass % HCl and heated at 120 °C in thick wall Teflon bottles for 16 hrs before the measurements according to a previous report,(Text Ref. 72) in order to fully digest the ds-DNA molecules in the samples for a more accurate phosphorous measurement. The bottles were weighted before and after heating to measure the mass loss due to evaporations (< 0.2%). Different dissolution concentrations (from 4 to 80 mg/L) were used for the samples in order to make sure that the signals of different elements all fall within the detection limits. The solutions were introduced into the spectrometer by a cross-flow nebulizer and Scott spray chamber, with a sample uptake rate of 2 mL/min and argon gas flow. The calibration spike solutions for Ca, P, Mg and Na were prepared from the standard ICP solutions (VWR), respectively. The possible disturbance of ds-DNA to the plasma was tested for solutions with different ds-DNA concentrations, and no influence was detected for up to 1 g/L of ds-DNA.

4. The text is revised accordingly

On Page 3, Line 78-82: The combination of 2D and 3D high resolution cryoTEM with FTIR, pH and [Ca²⁺] measurements, DLS/zeta potential measurements, inductively coupled plasma optical emission spectrometry (ICP-OES) as well as liquid- and solid-state NMR spectroscopy provides unique insights into the microscopic structure, transformation mechanism, and the liquid-like nature of PILP.

On Page 7, Line 157-159: The ds-DNA used (from salmon sperm, the same as in our original work) contained a small amount of Mg (15.25 mg/g, see ICP-OES results in Supplementary Table S2), leading to a Mg²⁺ concentration of ~1.6 mM in the crystallization solution.

On Page 14, Line 310-316: ICP-OES measurements indicated that the phosphorous mass fraction of the minerals reduced from 18.57 mg/g for the 2 h sample to 4.52 and 3.77 mg/g in the 3 h and 5 h samples, respectively (see Supplementary Table S2), indicating that most of the ds-DNA entrapped in the ACC is released during the transformation to the crystalline state (Text Ref. 59), and that the remaining ds-DNA becomes occluded in the calcite crystal. A similar change of phosphorous ratio was also detected by EDS measurements (see Supplementary Table S3).

Reviewer 3:

Reviewer comments (General):

This paper brings us much closer to understanding the microscopic structure of the PILP phase. Using cryoTEM, liquid and solid-state NMR, as well as DLS, ATR-FTIR, and conventional microscopies, the authors provide a thorough and in-depth characterization of the early stages of PILP, and provide a unifying theory of its structure. This type of characterization has been lacking in the community, and I am grateful to the authors for providing us with such a clear picture of a very useful but poorly understood, until now, method for obtaining calcium carbonate structures with a range of non-equilibrium morphologies. The characterization is performed with attention to detail and the resulting conclusions regarding the formation mechanism are well-reasoned and supported by experimental data. For these reasons, I recommend publication in Nature Communications after minor revisions. Below are specific comments for the authors to consider when revising their manuscript.

Author reply: We appreciate the reviewer for the very positive comments. We have revised our manuscript following the suggestion of the reviewer.

Reviewer comments 1:

It would be helpful to have a clear and concise summary of the criteria for liquid vs. nanoparticulate phases for the non-expert reader who hasn't followed the years of PILP debate and discussion. Such a summary could go into the end of the introduction or beginning of the results section. Something along the lines of what characteristics in TEM, NMR, etc. would indicate liquid vs. nanoparticulate composite. The authors scatter these criteria throughout the manuscript, and I think it would be good to put them all into one paragraph early in the manuscript.

Author reply: We agree with the reviewer on this suggestion.

Revisions made to the manuscript:

1. The following literature is added into the reference list as Ref. 32:

Zeng F, Tong Z, Feng H. N.m.r. investigation of phase separation in poly(N-isopropyl acrylamide)/water solutions. *Polymer* **38**, 5539-5544 (1997).

2. We have added a paragraph about these criteria in the introduction:

On Page 7, Line 43-47: Direct evidence for the liquidity of PILP could only be obtained by characterizations in hydrated status, such as cryoTEM or liquid state nuclear magnetic resonance (NMR) spectroscopy. The PILP droplets are expected to be continuous objects with smooth edges under cryoTEM (see also supplementary Fig. S1).(Text Ref. 21,30,31) Furthermore, the liquid-liquid phase separation should be reflected by extra peaks in liquid state NMR measurements.(Text Ref. 32)

Reviewer comments 2:

New data regarding the growth of calcite nanorods into track-etched membranes in the absence of additives is introduced in the Discussion section (Fig. S12). Please introduce all new results in the results section. I also suggest emphasizing that the reason the track-etch membrane results with and without additives are presented is because the ability to infiltrate pores and form non-equilibrium shapes has

been attributed to the liquid-like nature of PILP (see comment #1 – if the track etch membrane experiments are motivated as one of the observations that needs explanation, it will make more sense why they are included.).

Author reply: We fully agree with the reviewer on this suggestion.

Revisions made to the manuscript:

1. The original supplementary Fig. S10 is now moved up as Fig. S9 and described in the Results section as follows:

On Page 6-7, Line 140-142: The earliest products were observed after ~150 min of growth, which are nanoparticles (NPs) with diameters of ~50 nm (Fig. 2a, see **Supplementary Fig. S9-10** for the cryoTEM images of even earlier stages).

2. The original supplementary Fig. S11 and S12 are moved up as Fig. S4 and S6, respectively, and described in the Results section as follows:

On Page 4-5, Line 108-110: Under the same experimental conditions, ~20 μm sized calcite crystals were formed within 30 mins without the presence of pAsp (Supplementary Fig. S4), showing the ability of pAsp to inhibit the crystallization and stabilize ACC phase.

On Page 5, Line 114-116: Without the presence of pAsp, however, only small amount of short CaCO₃ nanorods are formed within the track etch membrane (Supplementary Fig. S5 a and d), which shows crystal-facet on their tips (Supplementary Fig. S6).

3. We have also emphasized that the formation of the non-equilibrium shapes are attributed to the liquid-like nature of PILP in the introduction:

On Page 2, Line 37-39: The formation of these non-equilibrium morphologies have been attributed to the liquid-like nature of PILP, being able to wet the solid substrates, or to be capillarity absorbed into the nanopores.(Text Ref. 20,23)

Reviewer comments 3:

For both the ATR-FTIR and ss-NMR experiments, the authors need to provide more details regarding sample preparation. Are solid precipitates isolated by centrifugation? Are KBr pellets made? Was the solid packed into the NMR tubes? Such sample preparation details are critical to understanding the results.

Author reply: We thank the reviewer for the kind suggestion.

Revisions made to the manuscript: We have added the information to the Methods part:

On Page 22, Line 553-556: Solid state ³¹P and ¹³C NMR spectra were recorded on a 9.4 T solid-state Varian NMR system (VNMRs) using a Varian 3.2 mm T3-HXY MAS probe, configured in double-resonance mode for ¹H-³¹P, respectively ¹H-¹³C. The samples (except ds-DNA powder, which was measured as it is) were isolated by centrifugation, dried at room temperature, and directly packed into the rotor without further treatment.

On Page 23, Line 572-574: Attenuated total reflection Fourier transform infrared (ATR-FTIR) spectra of the ds-DNA powder or centrifuge-separated and room temperature dried precipitates were obtained directly on the samples using a Varian FT-IR 3100 Spectrometer with Golden Gate ATR accessory

Reviewer comments 4:

On page 17, line 388, please provide a reference for the “traditional polyelectrolyte-enhanced wetting of surfaces.”

Author reply: We thank the reviewer for pointing this out. As discussed following the specific comment 2 of reviewer 1, the ability of PILP to attach to different surfaces should not be only determined by the polyelectrolytes. Furthermore, it is misleading to use the term “wetting” again since PILP is not a liquid phase and thus could not really wet surfaces, but only attach to them.

Revisions made to the manuscript:

We have corrected the statement into “the attachment of polyelectrolyte-stabilized colloidal particles to surfaces” on Page 18, Line 432, and has added the following literature as Ref. 66:

Böker A, He J, Emrick T, Russell TP. Self-assembly of nanoparticles at interfaces. *Soft Matter* 3, 1231-1248 (2007).

Reviewer comments 5:

On page 7, line 154, please specify in which figure the ds-DNA macromolecules can be seen.

Author reply: We thank the reviewer for the suggestion.

Revisions made to the manuscript:

We have modified the text accordingly:

On Page 7, Line 166, In this case however, also the macromolecules (ds-DNA, highlighted by yellow arrows in Fig. 2d) could be observed, intertwined with the ~2 nm subunits and protruding into the solution.

Reviewer comments 6:

In the Methods section, please provide an item number for the track-etch membranes used from VWR. These membranes are notoriously supplier and item number specific, especially as it relates to the surface treatments, which can vary. More information is required to repeat the experiments as described.

Author reply: We thank the reviewer for the suggestion.

Revisions made to the manuscript:

We have added the information to the Methods part:

On Page 20, Line 471-472, 10 µm thick poly-carbon track etch membranes with 50 nm (item No. 515-2026, supplier No. 110603) and 200 nm (item No. 515-2029, supplier number: 110609) sized pores were ordered from VWR.

Reviewer comments 7:

Units:

a. Throughout the manuscript, lowercase l's are used for liter instead of uppercase L's. (e.g., g/L instead of g/L)

b. Units are not provided for molecular weights. Please specify g/mol

Author reply: We thank the reviewer for pointing this out.

Revisions made to the manuscript:

We have modified the units accordingly throughout the entire manuscript and supplementary information.

Other corrections:

1. We have fixed a typo problem, where “solution” should be “aqueous solution”:

On Page 17, Line 385-386, Solid ACC NPs have a continuous structure, and usually dissolve within several minutes in aqueous solution.(Text Ref. 61)

2. The liquid state ^{31}P NMR spectra were recorded with trimethyl phosphate (TMP) as a reference. This information was missing in the previous version and is now added into the Methods part:

On Page 22, Line 551-552, spectra were recorded at 25 degrees accumulating 256 scans with recycle delay of 5 s, and referenced with respect to trimethyl phosphate (TMP).

3. The term “dynamic interactions” used in the discussions of ^{13}C CP-MAS SS-NMR results below supplementary Fig.S16 was not clear and has been corrected:

On Page 20, Line 323-324 of Supplementary Information: The different CP efficiency of ds-DNA and ACC indicates that local dynamics exist between the two components even in the solid-state, which influence the CP transfer for the ACC.

4. The following information is added into the Acknowledgements:

On Page 30, Line 844-846: A.A is an Awardee of the Weizmann Institute of Science -National Postdoctoral Award Program for Advancing Women in Science.

On Page 30, Line 847-848: Adelheid Elemans-Mehring and Emiel J.M. Hensen (Eindhoven University of Technology, The Netherlands) for help with ICP-OES measurements.

5. The descriptions of the supplementary Movies were not displayed correctly. They are now uploaded in separate documents.

Reviewers' comments:

Reviewer #1 (Remarks to the Author):

While I appreciate the time and efforts the authors have put to address my comments, I would have more appreciated if they made answers more concise and straight to the point.

I can see the authors have really tried to find the existence of 2 nm building units, obviously they failed to find any evidence. As this is one of the main claims of this work, I am hugely disappointed by this result. I am much sceptical with authors' original claim now. Authors' reasoning is low concentrations and instability of the clusters. This is highly unlikely to me. Their previous works have shown "(Pre)nucleation clusters" or "ion pair" in a few nm size without polymers, which are to be much unstable and short-lived. Typically polymers are known to stabilise precursors to extend their lifetimes ?

So I think the authors' claim should be softened, as indeed there is NO evidence of 2 nm building units regardless they are liquid or solid. By all means, the earliest and smallest species shown in this work are 30 nm -sized particles, which could be assemblies of smaller clusters, but equally could be chain-like structures or ACC with channels which have been proposed previously.

"without being stabilized by the polymers and assembled into the >30nm sized assemblies, these ACC clusters will immediately dissolve due to their high surface area ratio and the instability of ACC phase in water solution, leading to an extremely low equilibrium concentration of the individual clusters or smaller assemblies in our experiments. "

Unfortunately, current work doesn't support this statement. It should be stated as only speculation until further work will confirm.

I found that sometimes, authors' claims are contradicting one another. For example,

"Hence the ACC thin films should dissolve at the same time when the calcite/vaterite platelets form, especially around the front of crystallization where ACC is being faster consumed"....

First of all, Evidences of major amount of polymers remained in crystalline phases have been reported?

Is the main idea of PILP is based on aggregation-based crystallization? and the fact that polymer stabilizes ACC and inhibits from dissolution ? that is what the schematic is exactly implying, particularly the rod formation in membrane. If authors argue otherwise, the schematic should reflect that point of view as well.

I did ask "Out of curiosity, "nanoscopic fluidity" has ever been directly observed by in-situ EM ?"

The references the authors provide only with much larger liquid droplets 20-100 nm or solid nanoparticles. I assume then, there aren't reports of behavior of LIQUID droplets in 2-3 nm. This could be the case that the quick assembly to larger particles would be due to their liquidity?

Other than that, I am happy with authors' rigorous answers, and very thank all the authors for giving me the opportunity to review this nice work.

If authors are willing to understate their argument, clearly stating that they don't have any direct evidence of 2 nm solid particle units, I am happy to agree for the publication in nature communication.

Reviewer #2 (Remarks to the Author):

Line 128-140: Cryo-TEM shown in Figure 2 was performed on CaCO₃ thin film formation process with 10 mM CaCl₂ in presence of 25 mg/L PASP. Did the authors also perform controls where there is no PASP (Calcite control)? Or with PASP and Ca but no ammonium carbonate source (examine the PASP/Ca complex)? Please discuss the potential limitations of cryo-TEM, especially does the sampling procedure represent the whole crystallization solution and the process will not affect the crystallography?

Literature has reported a seeded crystal growth experiment using precursor system. For example, an existing large calcite crystal was placed in a mineralization solution of CaCl₂ with PASP, it was found that calcite nanorods grew vertically on one plane of calcite. Can the nanoparticle mechanism mentioned in this manuscript explains the observed nano-rod growth on an existing calcite crystal? It would be great the authors discussed more the potential implication of observed mechanism to those reported in literature. <https://pubs.acs.org/doi/abs/10.1021/cm035161r>

The data the authors presented seemed convincing. However, please discuss reference 23 and compare with the conclusion you just drew. If indeed the ACC (CaCO₃ stabilized with polymer) is actually nanoscopic particles only, how can the nanoparticles formed a singlecrystalline structure. It seems more convincing to start from a liquid-like feature to form a single crystalline structure. The particle fusion mechanism may easily lead to a polycrystalline structure, instead of single crystals.

Reviewer #3 (Remarks to the Author):

The authors have thoroughly addressed all previous comments. The revised manuscript is significantly improved as a result. Thank you!

Response to Reviewers' comments:

Reviewer 1:

Reviewer comments (General): While I appreciate the time and efforts the authors have put to address my comments, I would have more appreciated if they made answers more concise and straight to the point.

Author reply: The authors thank the reviewer for the positive comments and will try to answer in a more concise and straight way this time.

Reviewer Comment 1: I can see the authors have really tried to find the existence of 2 nm building units, obviously they failed to find any evidence. As this is one of the main claims of this work, I am hugely disappointed by this result. I am much sceptical with authors' original claim now. Authors' reasoning is low concentrations and instability of the clusters. This is highly unlikely to me. Their previous works have shown "(Pre)nucleation clusters" or "ion pair" in a few nm size without polymers, which are to be much unstable and short-lived. Typically polymers are known to stabilise precursors to extend their lifetimes? So I think the authors' claim should be softened, as indeed there is NO evidence of 2 nm building units regardless they are liquid or solid. By all means, the earliest and smallest species shown in this work are 30 nm –sized particles, which could be assemblies of smaller clusters, but equally could be chain-like structures or ACC with channels which have been proposed previously.

Author reply: We appreciate the reviewer for this critical and rigorous comment. Indeed, we were not able to detect individual ~2 nm sized clusters in the surrounding solution. At the same time we report that the subunits within the PILP particles are interconnected. We therefore fully agree with the reviewer that the existence of 2 nm building units should not be described as a direct observation in the

experimental part. We have modified the manuscript accordingly (see below), and clearly stated that this is only a proposal based on our experimental results.

Nevertheless, we still believe the “assembly of 2 nm clusters” is the most reasonable pathway of how the PILP particles are formed, considering the nanoparticulate morphology of the particles, and the uniform ~2 nm size of the subunits (although already interconnected) observed for all the four PILP systems we studied. With regards to the different concerns raised by the reviewer, we would like to emphasize that:

1. Most of our previous studies on clusters were performed in CaP systems (e.g. Habraken et al., *Nat. Commun.* 2013, Dey. et al., *Nat. Mater.* 2010, Nudelman et al., *Nat. Mater.* 2010), which has been widely reported to form nanosized clusters (Ref: Posner and Betts, *Acc Chem Res*, 1975), and is not expected to be same as the CaCO₃ system. Our only previous report on CaCO₃ clusters is the 2009 Science paper (Ref: Pouget et al., *Science*, 2009), which was performed using a very different experimental setup (“Kitano method” directly on a cryoTEM grid). Furthermore, we have recently realized and reported that the defocus value we used for that work was too large to confirm the existence of ~0.6 nm sized clusters considering the corresponding contrast transfer function (CTF) (Text Ref. 30: Smeets et al., *PNAS*, 2017).

In other words, so far we have not made reliably observations of CaCO₃ clusters which indeed may indicate that they are very unstable in aqueous solution. Although polymers are indeed able to stabilize ACC in aqueous solution, as was also shown in this study, there are no reports demonstrating that polymers could stabilize separated, individual clusters of CaCO₃ in aqueous environment. The fact that we do not visualize individual CaCO₃ clusters in our current study does therefore not contradict any previous report.

2. For the chain-like structure of ACC, we assume the reviewer is referring to the “dynamically ordered liquid-like oxyanion polymer (DOLLOP)” proposed by Demichelis et al. (*Nat. Commun.* 2011). The existence of these structures in PILP systems is possible, but we find no evidence in our cryoTEM experiments. Also in this case the polymeric chains would be heavily hydrated, which is not in line with the dense appearance of the ~2 nm nanoparticulate textures (for the density comparison see also Text Ref. 30: Smeets et al., *PNAS*, 2017). For the ACC with channels, we suspect the reviewer refers to the model proposed by the Reeder group (Goodwin et al., *Chem. Mater.*, 2010), which suggests there are interconnected Ca-poor channels in ACC that are ~1 nm in diameter. This is however a model proposed for ACC synthesized without polymers, and these channels were invisible in our previous cryoTEM studies on ACC formed in absence of polymer (Text Ref. 62: Pichon et al., *J. Am. Chem. Soc.*, 2008.). Most of all, it is a description of the ACC morphology rather than its formation pathway. To prevent confusion, we decide to not involve these two studies in our discussions.

Revisions made to the manuscript:

1. All the descriptions that the PILP nanoparticles are directly observed to be assembled from ~2 nm subunits are corrected:

- 1) **Line 17-18:** The initial products are 30-50 nm amorphous calcium carbonate (ACC) nanoparticles with ~2 nm nanoparticulate texture.
- 2) **Line 20:** Our results suggest that “PILP” is actually a polymer-driven assembly of ACC clusters
- 3) **Line 142-143:** High magnification images showed that the NPs appear to consist of assemblies of ~2 nm subunits
- 4) **Line 167-168:** which are near-identical to those formed in the presence of pAsp, and also appear to be assembled from ~2 nm subunits
- 5) **Line 173-174:** Similar NPs with ~2 nm nanoparticulate textures were found in the early stages of the pAA or pAH (25 mg/L) induced PILP processes.
- 6) **Line 331-332:** In fact, PILP consists of ACC nanoparticles with a ~2 nm nanoparticulate texture, and no other product was detected before the formation of this phase in our experiments (see Supplementary Fig. S9-10).
- 7) **Line 342:** Our proposal of that PILP is assembled from ~2 nm sized ACC clusters
- 8) **Line 424-425:** In conclusion we show here that the CaCO₃ PILP consists of gel-like nanoparticles, which should be formed by the assembly of polymer stabilized ~2 nm ACC clusters.
- 9) **Supplementary Information, Line 220-221:** showing the ~2 nm sized nanoparticulate texture of the NPs
- 10) **Supplementary Information, Line 224:** ~2 nm sized nanoparticulate texture was observed for the NPs
- 11) **Supplementary Information, Line 236-237:** CryoTEM shows the NPs possess ~2 nm sized nanoparticulate texture (Fig. S13c). Similar structure was also found in a recent TEM study of a pAA/ACC hydrogel.
- 12) **Supplementary Information, Line 240-241:** Zoom-in image of the interface between NPs showed the ~2 nm sized nanoparticulate texture of the NPs
- 13) **Supplementary Information, Line 273:** The PILP NPs with the ~2 nm sized nanoparticulate texture were thus hidden

2. A discussion is added to make it clear that the “assembly of ~2 nm nanoclusters” is not a direct observation, but a proposal based on the experimental results:

Line 333-336: The same ~2 nm nanoparticulate texture was found in all the four PILP systems we studied, which strongly suggests that PILP is assembled from ~2 nm clusters. However, these were not detected as individual clusters in the early stage reaction solutions, which may mean that they are not stable before assembling on the polymers.

Reviewer Comment 2:

“without being stabilized by the polymers and assembled into the >30nm sized assemblies, these ACC clusters will immediately dissolve due to their high surface area ratio and the instability of ACC phase in water solution, leading to an extremely low equilibrium concentration of the individual clusters or smaller assemblies in our experiments. “

Unfortunately, current work doesn't support this statement. It should be stated as only speculation until further work will confirm.

Author reply: We agree with the reviewer on this comment.

Revisions made to the manuscript:

1. We have removed the statement at this point and toned it down in the discussion in **Line 333-336:** The same ~2 nm nanoparticulate texture was found in all the four PILP systems we studied, which strongly suggests that PILP is assembled from ~2 nm clusters. However, these were not detected as individual clusters in the early stage reaction solutions, which may mean that they are not stable before assembling on the polymers.

(see also the reply to the comment 1 of reviewer)

2. A similar statement in the **Supplementary Information** is also corrected accordingly in **Line 167-168:** The extremely low concentration of individual clusters or smaller assemblies suggests that they are not stable before assembling on the polymers.

Reviewer Comment 3:

I found that sometimes, authors' claims are contradicting one another. For example, “Hence the ACC thin films should dissolve at the same time when the calcite/vaterite platelets form, especially around the front of crystallization where ACC is being faster consumed”.... First of all, Evidences of major amount of polymers remained in crystalline phases have been reported? Is the main idea of PILP is based on aggregation-based crystallization? and the fact that polymer stabilizes ACC and inhibits from dissolution ? that is what the schematic is exactly implying, particularly the rod formation in membrane. If authors argue otherwise, the schematic should reflect that point of view as well.

Author reply: We are not sure about the reviewer's concern here, and respectfully suggest there may be some misunderstanding. We want to emphasize that on the right side of Figure 6B, we are only discussing the ACC stage of rod formation in the membrane.

As we have stated in the manuscript (e.g., Line 296-297), most of the polymer is excluded during the crystallization process, and only a small amount is trapped in the crystals. And we only observed the aggregation of amorphous PILP particles, but not crystalline particles. The polymers indeed stabilizes ACC, which we do not think is contradicting with any other of our claims.

Reviewer Comment 4:

I did ask “Out of curiosity, “nanoscopic fluidity” has ever been directly observed by in-situ EM ?”
The references the authors provide only with much larger liquid droplets 20-100 nm or solid nanoparticles. I assume then, there aren't reports of behavior of LIQUID droplets in 2-3 nm. This could be the case that the quick assembly to larger particles would be due to their liquidity?

Author reply: Indeed, as far as we know, there is no EM report on 2-3 nm liquid droplets. However, we do not think the clusters in our experiments are liquid droplets, since they do not coalesce into a bigger continuous structure. Also their contrast in cryoTEM visualizations is significantly higher than the surrounding solution, which is not expected for a highly hydrated CaCO_3 liquid precursor (see also Text Ref. 30: Smeets et al., *PNAS*, 2017). At least it appears that the clusters have already lost their liquidity when they form the assemblies.

Reviewer comments (Conclusion):

Other than that, I am happy with authors' rigorous answers, and very thank all the authors for giving me the opportunity to review this nice work. If authors are willing to understate their argument, clearly stating that they don't have any direct evidence of 2 nm solid particle units, I am happy to agree for the publication in nature communication.

Author reply: The authors would like to thank the reviewer again for carefully reading our manuscript and making the very inspiring and constructive comments. We have modified the manuscript following his/her suggestions, and hope it is now ready for the publication in Nature Communications.

Reviewer 2:

Reviewer Comment 1:

Line 128-140: Cryo-TEM shown in Figure 2 was performed on CaCO₃ thin film formation process with 10 mM CaCl₂ in presence of 25 mg/L PASP. Did the authors also perform controls where there is no PASP (Calcite control)? Or with PASP and Ca but no ammonium carbonate source (examine the PASP/Ca complex)? Please discuss the potential limitations of cryo-TEM, especially does the sampling procedure represent the whole crystallization solution and the process will not affect the crystallography?

Author reply: We thank the reviewer for these suggestions. A systematic cryoTEM study of CaCO₃ mineralization using ammonium carbonate diffusion method without presence of pAsp has already been performed by us previously, under the same CaCl₂ concentration (10 mM), which was the Text Ref. 62 (now 47): Pichon et al., *J. Am. Chem. Soc.*, 2008. In this previous study ACC nanoparticles without fine structure were formed, and crystallization happened within 5 mins, which clearly contrast with the PILP process. This study is now also referred to in the description of Figure 2.

For the solution with pAsp and Ca, no product could be found without ammonium carbonate diffusion. We have made this now more clear in the Supplementary Information.

With regards to the limitations of cryoTEM as a technique, there have been abundant discussions on the pros and cons, which are out of the scope of the current manuscript. As Text Ref. 31 (Dubochet J, et al. *Q. Rev. Biophys.*, 1988) and 41 (Patterson et al., *Acc. Chem. Res.*, 2017) are nice examples of these discussions we have included these also in the Method section pointing specifically to this issue.

Revisions made to the manuscript:

1. Previous Ref. 62 is now shifted forwards as Ref. 47, and referred to in **Line 151-153:**

The results clearly contrast with our previous cryoTEM study that used similar reaction conditions but without the presence of polymers, Text Ref. 47 where ACC nanoparticles without fine structure were formed, and crystallization happened within 5 mins.

2. The following statement in the **Supplementary Information** has been modified in **Line 135-136:**

In the experiments with 25 mg/L of pAsp, no liquid-like droplets or any other product was observed by cryoTEM in the solution before the NPs shown in Fig. 2a.

3. The following text was added to the Method section: Cryogenic Transmission Electron Microscopy, in **Line 492-493:**

For the discussions on the limitations of cryoTEM please see Ref. 31 and 41.

Reviewer Comment 2:

Literature has reported a seeded crystal growth experiment using precursor system. For example, an existing large calcite crystal was placed in a mineralization solution of CaCl₂ with PAsP, it was found that calcite nanorods grew vertically on one plane of calcite. Can the nanoparticle mechanism mentioned in this manuscript explain the observed nano-rod growth on an existing calcite crystal? It would be great if the authors discussed more the potential implication of the observed mechanism to those reported in literature. <https://pubs.acs.org/doi/abs/10.1021/cm035161r>

Author reply:

The authors assume the reviewer is referring to nano-rod formation with pAA (instead of pASP), which was described in the literature he/she mentioned. Those nano-rods are formed in bulk solution without the presence of a track-etch membrane, which are different from the nano-rod formation within track-etch membrane that we described. The formation mechanism for those nanorods was proposed to be similar to the vapor-liquid-solid (VLS) or solution-liquid-solid (SLS) mechanism used for semiconductor nanorod synthesis. This phenomenon has been reported for a few PILP systems, e.g., the ones using pAA or pAH (see also Text Ref. 21: Cantaert et al., *Adv. Funct. Mater.*, 2012) with or without the presence of seeded crystals, but is not universal for all the PILP processes. Furthermore, nanorod formation without the presence of track etch membrane was not observed in our experiments even when pAA or pAH was used, suggesting the process may be highly sensitive to detailed experimental parameters (e.g., ammonium carbonate diffusion speed). As a result, we decide to not further discuss the formation pathway of those nanorods. Meanwhile, to prevent possible misunderstanding, we have made it clear in our manuscript that the nanorods we described are different from those mentioned by the reviewer, and we did not observe nanorod formation without the presence of track etch membranes in this study.

Revisions made to the manuscript:

1. The following reference is added into the manuscript as Ref. 65:

Olszta MJ, Gajjeraman S, Kaufman M, Gower LB. Nanofibrous Calcite Synthesized via a Solution–Precursor–Solid Mechanism. *Chem Mater* **16**, 2355-2362 (2004).

2. A statement has been added in **Line 390-393** to make it clear that the nanorods we discussed are different from those formed in bulk solution:

It is noteworthy that the nanorods we described are different from the CaCO₃ nanorods formed in the bulk solution, as has been reported for pAA and pAH induced PILP processes. REF21 and 65 Those nanorods are however not observed in our experiments, even when pAA or pAH was used, and their detailed formation mechanism so far remains unresolved.

Reviewer Comment 3:

The data the authors presented seemed convincing. However, please discuss reference 23 and compare with the conclusion you just drew. If indeed the ACC (CaCO₃ stabilized with polymer) is actually nanoscopic particles only, how can the nanoparticles formed a singlecrystalline structure. It seems more convincing to start from a liquid-like feature to form a single crystalline structure. The particle fusion mechanism may easily lead to a polycrystalline structure, instead of single crystals.

Author reply:

The authors respectfully disagree with the reviewer on this suggestion. The aggregation of ACC nanoparticles has been shown to lead to single crystalline calcite that maintain the nanoparticulate texture both in *vitro* and *vivo*. (See Text Ref. 38—Gal et al., *Adv. Funct. Mater.*, 2014 and also this Ref: Politi et al, *Science*, 2004.) Also in the Text Ref. 23 mentioned by the reviewer (Kim et al., *Angew. Chem. Int. Ed.*, 2011), single crystal formation was not correlated to a liquid precursor, although it was proposed that a liquid precursor may promote the infiltration process. As these authors suggested, single crystal formation may be due to limited crystal nucleation within the pores of track etch membrane, and/or by a competitive crystal growth mechanism which lead to a dominant crystalline growth at the expense of smaller ones.

Therefore, we do not feel there is any conflict between our results and previous reports. Meanwhile, our current study mainly focuses on the microscopic structure of the PILP phase, and discussions on the single crystal formation mechanism are not in the heart of our study, and would merely complicate the story. So we prefer not to include this discussion in the present paper.

Reviewer 3:

Reviewer comments (General): The authors have thoroughly addressed all previous comments. The revised manuscript is significantly improved as a result. Thank you!

Author reply: The authors appreciate the reviewer for the very positive comment, and hope to thank the reviewer for his/her help on improving the quality of our manuscript.

Other corrections:

1. Statements on data and code availabilities have been added into the Method section in **Line 587-592**:

Code Availability

All the Matlab scripts used for the morphological processing and analysis of tomographic results are available from the corresponding authors upon reasonable request.

Data availability

All the data that support the findings of this study are available from the corresponding authors upon reasonable request.

2. Several typo mistakes are corrected, which are not listed here but highlighted in the text.

REVIEWERS' COMMENTS:

Reviewer #1 (Remarks to the Author):

i am happy with their changes, and would like to recommend to be published in nature communications.

Reviewer #2 (Remarks to the Author):

The authors have diligently revised the manuscript and it has been improved significantly during the process. This article, after publish, will be controversial but the reviewer think it will help the field move forward and get more researchers' interests.

One critique I have is the different interpretation of other researcher's results and the authors' own results. The author claimed there is no liquid-liquid phase separation, however they did observe there is separation of ACC phase and crystal phase during the observation of the process. The pioneer, Dr. Laurie Gower, who first observed this CaCO₃ crystallization phenomena in the presence of PAsp in the 90s and hypothesize there is liquid-liquid like phase separation. Aren't the liquid in Dr. Gower's paper not referring to polymer stabilized mineral phases which behave like liquids macroscopically? Otherwise, what her paper refers to as the "liquid" phase? The reviewer feel like the authors just need to present results as he/she observed and due to the partial uncertainty nature of the smaller units nanoparticles and the possible limitation of cryo-TEM, there is no need to start another phrase such as Polymer-induced-nanoparticle-precursor (PINP). The PINP might be correct under microscopic scale (but may be incorrect who knows), but the original PILP may reflect the whole nature of this unique process, where the liquid can be nanodroplets, fluidic nanoparticles clusters, or something else. In essence, the field really welcomes more research into this area and getting to the bottom of the truth may take more time and resources. So for continuity of research and respect of others' work, unless the others' theory or experiment is wrong, there is no need to use a new term which may not truly represent the process.

Response to Reviewers' comments:

Reviewer 1:

Reviewer comments (Conclusion): i am happy with their changes, and would like to recommend to be published in nature communications.

Author reply: The authors appreciate the reviewer for the publication recommendation.

Reviewer 2:

Reviewer comments (Conclusion): The authors have diligently revised the manuscript and it has been improved significantly during the process. This article, after publish, will be controversial but the reviewer think it will help the field move forward and get more researchers' interests.

Author reply: The authors appreciate the reviewer for the very positive comment, and would be extremely happy if our work could benefit the development of this research field.

Reviewer comments (Remark): One critique I have is the different interpretation of other researcher's results and the authors' own results. The author claimed there is no liquid-liquid phase separation, however they did observe there is separation of ACC phase and crystal phase during the observation of the process. The pioneer, Dr. Laurie Gower, who first observed this CaCO₃ crystallization phenomena in the presence of PAsp in the 90s and hypothesize there is liquid-liquid like phase separation. Aren't the liquid in Dr. Gower's paper not referring to polymer stabilized mineral phases which behave like liquids macroscopically? Otherwise, what her paper refers to as the "liquid" phase? The reviewer feel like the authors just need to present results as he/she observed and due to the partial uncertainty nature of the smaller units nanoparticles and the possible limitation of cryo-TEM, there is no need to start another phrase such as Polymer-induced-nanoparticle-precursor (PINP). The PINP might be correct under microscopic scale (but may be incorrect who knows), but the original PILP may reflect the whole nature of this unique process, where the liquid can be nanodroplets, fluidic nanoparticles clusters, or something else. In essence, the field really welcomes more research into this area and getting to the bottom of the truth may take more time and resources. So for continuity of research and respect of others' work, unless the others' theory or experiment is wrong, there is no need to use a new term which may not truly represent the process.

Author reply: We understand and agree with the reviewer's concern about the continuity of the research and respect of the previous work in this field. Indeed, although our work indicate that PILP is not really a liquid phase, there is no need to start a new term for this system before we can fully understand it. We therefore decide to change our closing remark, now only to say that the 'polymer-induced liquid precursor' in fact is a 'polymer-induced liquid-like precursor', without changing the abbreviation PILP.

Revisions made to the manuscript: We have revised the closing remark back into: "We therefore suggest to use the abbreviation PILP for polymer-induced liquid-like precursor, acknowledging the exceptional macroscopic properties of these polymer directed crystallization systems."